



**Biogeochemical Impact of Cable Bacteria on Coastal Black Sea Sediment**
**Martijn Hermans[1], Nils Risgaard-Petersen[2,3], Filip J.R. Meysman[4,5] and Caroline P. Slomp[1]**
[1]Department of Earth Sciences, Faculty of Geosciences, Utrecht University, P.O Box 80021, 3508 TA
Utrecht, the Netherlands
[2]Center for Geomicrobiology, Section for Microbiology, Department of Bioscience, Aarhus
University, Aarhus, Denmark
[3]Center for Electromicrobiology, Section for Microbiology, Department of Bioscience, Aarhus
University, Aarhus, Denmark
[4]Center of Excellence for Microbial Systems Technology, Department of Biology, University of
Antwerp, Wilrijk, Belgium
[5]Department of Biotechnology, Delft University of Technology, Delft, the Netherlands
**ABSTRACT**
Cable bacteria can strongly alter sediment biogeochemistry. Here, we used laboratory
incubations to assess whether cable bacteria can establish in iron (Fe) monosulphide-poor coastal
Black Sea sediment and to determine the impact of their activity on the cycling of Fe, phosphorus (P)
and sulphur (S). Microsensor depth profiles of oxygen, sulphide and pH in combination with electric
potential profiling and FISH analyses showed a rapid development (<5 days) of cable bacteria,
followed by a long period of activity (>200 days). During most of the experiment, the current density
correlated linearly with the oxygen demand. Sediment oxygen uptake was attributed to activity of
cable bacteria and the oxidation of reduced products from anaerobic degradation of organic matter,
such as ammonium. Pore water sulphide was low (<5 μM) throughout the experiment. Sulphate
reduction acted as the main source of sulphide for cable bacteria. Pore water $Fe^{2+}$ reached levels of up
to 1.7 mM during the incubations, due to the dissolution of FeS (30%) and siderite, an Fe carbonate
mineral (70%). Following upward diffusion of $Fe^{2+}$, a surface enrichment of Fe oxides formed. Hence,
besides FeS, siderite may act as a major source of Fe for Fe oxides in coastal surface sediments where
cable bacteria are active. Using μXRF, we show that the enrichments in Fe oxides induced by cable





bacteria are located in a thin subsurface layer of 0.3 mm. We show that similar subsurface layers
enriched in Fe and P are also observed at field sites where cable bacteria were recently active and
little bioturbation occurs. This suggests that such subsurface Fe oxide layers, which are not always
visible to the eye, could potentially be a marker for recent activity of cable bacteria.
Key words: cable bacteria, elemental cycling, solute fluxes, iron

## 1. INTRODUCTION

Depletion of oxygen ($O_2$) in bottom waters of coastal areas is increasing worldwide, as a
consequence of eutrophication and climate change (Diaz and Rosenberg 2008; Schmidtko et al. 2017;
Breitburg et al. 2018). Low $O_2$ can lead to the development of coastal 'dead zones' characterised by
recurrent mortality of marine life (Rabalais et al. 2002; Diaz and Rosenberg 2008). Progressive
eutrophication induces a characteristic response of coastal systems with transient and seasonal
hypoxia ($O_2 < 63$ μM) transitioning into permanent anoxia ($O_2 = 0$ μM). In this later stage, free
sulphide ($H_2S$) may escape from the sediment and accumulate in the bottom water, a condition
referred to as euxinia (Diaz and Rosenberg 2008; Kemp et al. 2009; Rabalais et al. 2014). As $H_2S$ is
highly toxic to higher fauna, the development of euxinia may aggravate the ecological consequences.
However, the presence of iron (Fe) and manganese (Mn) oxides in surface sediments may delay this
transition towards euxinia by removing $H_2S$ and thus preventing an efflux of $H_2S$ to the overlying
water (Kristiansen et al. 2002; Kristensen et al. 2003; Diaz and Rosenberg 2008).
Cable bacteria are multicellular filamentous sulphur(S)-oxidising bacteria (Pfeffer et al. 2012)
that strongly enhance the formation of Fe and Mn oxides and efficiently remove $H_2S$ from surface
sediments (Risgaard-Petersen et al. 2012; Seitaj et al. 2015; Sulu-Gambari et al. 2016a). Cable
bacteria belong to the *Desulfobulbaceae* family of the Deltaproteobacteria (Trojan et al. 2016;
Kjeldsen et al. 2019). Cable bacteria can spatially link the oxidation of $H_2S$ in deeper sediments to the
reduction of $O_2$ near the sediment-water interface by transporting electrons over centimetre scale
distances (Pfeffer et al. 2012) through a conductive fibre network that is embedded in the cell
envelope (Meysman et al. 2019). This spatial coupling of surficial $O_2$ reduction with $H_2S$ oxidation at



several centimetres depth provides cable bacteria a competitive advantage over other S-oxidising
bacteria in aquatic environments (Meysman 2018). Cable bacteria have been documented in a range
of fresh water (Risgaard-Petersen et al. 2015; Müller et al. 2016) and marine environments (Malkin et
al. 2014; Burdorf et al. 2017), however, they appear to be particularly active in sediments overlain by
seasonally hypoxic bottom waters (Seitaj et al. 2015; Burdorf et al. 2018).

The metabolic activity of cable bacteria establishes an electrical circuit in the sediment, which

involves an electron current through the cable bacteria filaments (Bjerg et al. 2018), and an ionic
current through the pore water in the opposite direction (Naudet and Revil 2005; Revil et al. 2010;
Risgaard-Petersen et al. 2012). As a consequence, an electric potential (EP) is generated in the
sediment, which can be used as a reliable indicator for activity of cable bacteria (Risgaard-Petersen et
al. 2014).

Cable bacteria activity additionally generates a distinct biogeochemical signature, that can be

assessed by pH, $O_2$ and $H_2S$ depth profiling (Nielsen et al. 2010). Their activity leads to the
development of a suboxic zone (i.e. a zone where $O_2$ and $H_2S$ are both absent), and also induces a pH
profile that strongly changes with depth. Cathodic $O_2$ reduction ($O_2 + 4H^+ + 4e^- \rightarrow 2H_2O$) in the oxic
zone of the sediment results in a pH maximum (~9) due to proton consumption, whereas  anodic
sulphide oxidation ($H_2S + 4\ H_2O \rightarrow SO_4^{2-} + 10H^+ + 8e^-$) causes a pH minimum (<6.5) in the anoxic
zone (Fig. 1A; Nielsen et al. 2010; Meysman et al. 2015).

The presence of cable bacteria in sediments can strongly impact the elemental cycling of Fe,

Mn, Ca and S (Risgaard-Petersen et al. 2012; Seitaj et al. 2015; Rao et al. 2016; Sulu-Gambari et al.
2016a; van de Velde et al. 2016). Pore water acidification induced by cable bacteria activity can lead
to dissolution of calcium (Ca) carbonates, Fe carbonates (siderite), Mn carbonates and FeS in the zone
where the pH is low, thus generating high concentrations of $Fe^{2+}$ and $Mn^{2+}$ in the pore water
(Risgaard-Petersen et al. 2012; Rao et al. 2016). When these dissolved species diffuse upward this can
lead to strong enrichments of Fe and Mn oxides upon contact with $O_2$, or for dissolved $Fe^{2+}$, also upon
contact with Mn oxides (Wang and Van Cappellen 1996; Seitaj et al. 2015; Sulu-Gambari et al.





2016a). These metal oxides are capable of efficiently buffering the benthic release of $H_2S$ and
phosphate ($HPO_4^{2-}$) during periods with low bottom water $O_2$. This so-called 'firewall' for $H_2S$ and
alteration of the timing of $HPO_4^{2-}$ release linked to this buffering can play a key role in regulating
water quality in seasonally hypoxic coastal systems (Seitaj et al. 2015; Sulu-Gambari et al. 2016b;
Hermans et al. 2019a).

In coastal sediments, $O_2$ typically penetrates to a depth of only several mm's below the

sediment-water interface (Rasmussen and Jørgensen 1992; Rabouille et al. 2003; Glud 2008). This
also holds true for sediments inhabited by active cable bacteria (Nielsen et al. 2010; Pfeffer et al.
2012; Larsen et al. 2015). Hence, the oxidation of upward diffusing $Fe^{2+}$ and $Mn^{2+}$ is expected to take
place below and not at the sediment-water interface. We hypothesise that, as a consequence, in the
initial stages of cable bacteria activity and in the absence of bioturbation, most Fe and Mn oxide
enrichments will be restricted to a thin subsurface layer of the sediment. However, the sample
resolution and timing of the collection of solid phase data in field and laboratory studies published so
far do not allow an assessment of this hypothesis.

Cable bacteria are suggested to thrive in coastal sediments characterised by high rates of $H_2S$

production due to high rates of organic matter mineralisation (Malkin et al. 2014; Burdorf et al. 2017;
Hermans et al. 2019a). Laboratory and model studies have shown that the dissolution of FeS accounts
for 12 to 94% of the $H_2S$ consumed by cable bacteria, while the other source is $H_2S$ production from
the reduction of $SO_4^{2-}$ (Risgaard-Petersen et al. 2012; Meysman et al. 2015; Burdorf et al. 2018). At
present, it is not known if cable bacteria activity can establish in sediments that are relatively low in
FeS and dissolved $H_2S$.

In this study, we assess whether cable bacteria activity can establish in sediments that are

relatively poor in FeS in an incubation experiment using siderite-bearing sediments from a coastal site
in the Black Sea. The metabolic activity of cable bacteria is monitored using microsensor profiles of
pH, $O_2$, $H_2S$ and EP. We also use sediment Fe and P speciation and µXRF of resin-embedded
sediments to test whether we find evidence for subsurface enrichments in Fe oxides and associated P.





We find a rapid establishment of cable bacteria (<5 days) and the development of an Fe oxide-rich
subsurface layer, with the majority of the Fe ~70% supplied through dissolution of siderite induced by
cable bacteria activity. The depth of the Fe oxide layer was directly related to the $O_2$ penetration depth
and we propose that such subsurface enrichments in Fe, which also can contain P and Mn, can be used
as a marker for recent cable bacteria activity.

## 2. METHODS AND MATERIALS


### 2.1. Study Area and Experimental Set-up


In September 2015, 16 sediment cores (⌀10 cm) were retrieved at a coastal site on the north-
western shelf of the Black Sea (27 m water depth; Fig. 1B; Table 1) using a multicorer (Oktopus
GmbH, Germany) as described in Lenstra et al. (2019). The overlying water was discarded, and the
upper 10 cm of the sediment was transferred into nitrogen purged aluminium bags that were sealed
and stored at 4 °C for several months. The anoxic storage is expected to have led to the death of all
macrofauna and most meiofauna (Coull and Chandler 2001; Riedel et al. 2012). Prior to incubation,
the sediment was passed through a 4 mm sieve to remove large debris and homogenised.
Subsequently, the sediment was transferred to 18 transparent polycarbonate cores (⌀6 cm; 20 cm
length).
The bottom 15 cm of these cores was filled with sediment and the upper 5 cm with overlying
water. The cores were placed in two aquaria filled with artificial seawater (Instant Ocean Sea Salt +
Ultra High Quality (UHQ) water) with a salinity of 17.9, identical to the bottom water salinity at the
study site. The artificial sea water contained negligible concentrations of $NH_4^+$, $NO_3^-$, Fe, Mn and P as
described in Atkinson (1997) and Hovanec and Coshland (2004). The aquaria were kept in the dark at
a constant temperature (~20 °C), and the water was continuously aerated by two aquarium pumps.
Sixteen out of eighteen cores were exposed to oxygenated overlying water in the aquaria, whereas the
two remaining cores served as an anoxic control treatment. The control cores were tightly sealed with
rubber stoppers, to prevent the growth of cable bacteria by excluding $O_2$ (Nielsen et al. 2010).



Sampling for pore water and solid-phase analyses was performed at eight time points over a
total incubation period of 621 days. Each time point involved a three day procedure. On the first day,
microsensor depth profiles of EP, $O_2$, pH and $H_2S$ were obtained in two randomly selected oxic cores
and the two anoxic control cores ($O_2$ profiling was not performed in the anoxic cores). On the second
day, solute fluxes were measured in the same oxic cores that were used for microsensor depth
profiling on the previous day. On the third day, the two cores were sectioned, of which only one core
was processed further for pore water and solid-phase analyses. Photographs were taken at four time
points (day 12; 33; 170 and 621) from one oxic core to follow the visual development of the surface
sediment during the experiment.
**2.2. High-resolution Microsensor Depth Profiling**
High-resolution depth profiles of pH, $O_2$ and $H_2S$ were obtained (50-μm depth resolution; 3
replicate profiles per oxic core; 2 replicate profiles per anoxic core) using commercial micro
electrodes (Unisense A.S., Denmark). The $O_2$ sensor was re-calibrated prior to each measurement,
using saturated bottom water (100% [$O_2$]) and the deeper sediment horizons (0% [$O_2$]) as calibration
points. Calibrations of the pH and $H_2S$ electrodes were performed as described in Hermans et al.
(2019b). pH values are reported on the total scale. For depth profiling of EP (500-μm resolution; 3
replicates per core), micro electrodes were used that were custom built at Aarhus University, as
described in Damgaard et al. (2014). A robust reference electrode (Ref-RM, Unisense, A.S.,
Denmark) was used during EP and pH measurements. To exclude turbulence-induced variations in the
potential of the reference electrode during EP profiling, a silicon tube filled with foam was mounted
on the tip of the reference electrode.
**2.3. Solute Flux Measurements**
Solute flux incubations were performed for $NH_4^+$, $Fe^{2+}$, $Mn^{2+}$, $Ca^{2+}$, $HPO_4^{2-}$ and $H_4SiO_4$. At
each time point, one core was placed outside the aquarium, and the isolated volume of overlying
water in the core was continuously aerated. Parafilm was wrapped on top of the cores to prevent
evaporation. Water samples of 3 mL were retrieved at 7 time points over 24 hours. The same volume
of fresh artificial seawater was added to the cores directly after taking each sample. The samples were





filtered (0.45 μm), and subsamples were taken for ammonium (1 mL) and for metals (1 mL; acidified
with 10 μL Suprapur® HCl (35%) per mL sample), which were stored at -20°C and 4°C respectively
until further analysis.

### 2.4. Pore Water and Sediment Collection

At each time point, two cores were sectioned at 0.5-1 cm resolution with an UWITEC push-
up pole in a nitrogen-purged glovebag, but only samples for one core were used for sediment and pore
water collection and analyses. Bottom water samples were retrieved from the overlying water in the
cores. Slices for each depth interval were centrifuged at 3500 rpm for 20 minutes for pore water
retrieval. Samples (1 mL) for $NH_4^+$ were taken and stored at -20°C until analysis. Samples (1 mL) for
pore water S, Fe, Mn, Ca, P and Si were also collected and acidified with 10 μL Suprapur® HCl
(35%) per mL sample, which were stored at 4°C until analysis. Centrifuged sediment samples were
freeze-dried and ground to a fine powder in a nitrogen-purged glovebox under a strictly anoxic
environment to prevent oxidation (Kraal et al. 2009; Kraal and Slomp 2014). Only the top 5 cm of the
solid-phase samples were analysed in further detail. The porosity (Supporting Information 1.1; Table
S1) was calculated from the weight loss upon freeze-drying, using a sediment density of 2.65 g cm$^{-3}$
(Burdige 2006). Salt corrections were performed on the solid-phase data using the gravimetric water
content and salinity to determine the amount of salt after freeze-drying.

### 2.5. Chemical Analysis of the Water and Sediment

Concentrations of $NH_4^+$ in the pore water and solute flux samples were determined using the
phenol hypochlorite method (Koroleff 1969). The total Fe, Mn, Ca, P and Si concentrations (which
are assumed to represent $Fe^{2+}$, $Mn^{2+}$, $Ca^{2+}$, $HPO_4^{2-}$ and $H_4SiO_4$) in the pore water and solute flux
samples were determined using Inductively Coupled Plasma-Optical Emission Spectroscopy (ICP-
OES, Spectro Arcos). Dissolved Fe and Mn are assumed to be present in the form of $Fe^{2+}$ and $Mn^{2+}$,
however some $Mn^{3+}$ (Madison et al. 2013) or colloidal and nanoparticulate Fe and Mn might also be
available (Boyd and Ellwood 2010; Raiswell and Canfield 2012). Concentrations of P and S are
assumed to represent $HPO_4^{2-}$ and $SO_4^{2-}$ respectively. The colourimetric detection limit for $NH_4^+$ was





0.5 µM. The practical detection limit on the ICP-OES for Fe, Mn and P was 0.73, 0.11 and 7.30 µM,
respectively.

Solid-phase Fe was fractionated into [1] labile ferric Fe (hydr)oxides and ferrous Fe (FeS +

$FeCO_3$), [2] crystalline Fe minerals, [3] magnetite and [4] pyrite (Supporting Information 1.2; Table
S2), using a combination of two operational extraction methods (Poulton and Canfield 2005; Claff et
al. 2010) as described by Kraal et al. (2017). Concentrations of Fe in all extracts were determined
using the colourimetric phenanthroline method (APHA 2005). Solid-phase S was separated into [1]
acid volatile sulphur (AVS; representing FeS) and [2] chromium reducible sulphur (CRS; representing
$FeS_2$; Table S1) using the method after Burton et al. (2006; 2008) as modified by Kraal et al. (2013).
Sulphide released during the S extraction was trapped as ZnS in alkaline Zn acetate traps.
Concentrations of S were determined by iodometric titration (APHA 2005). Solid-phase siderite
($FeCO_3$) was determined by subtracting AVS from the labile ferrous concentrations retrieved from the
first step of the Fe extraction. Solid-phase P was fractionated into [1] exchangeable P, [2] citrate-
dithionite-bicarbonate (CDB)-P, [3] authigenic P, [4] detrital P and [5] organic P (Table S2) after
Ruttenberg (1992) as modified by Slomp et al. (1996). The sum of exchangeable P and CDB-P
represents metal bound P, as described in Hermans et al. (2019b). Concentrations of P in all extracts,
except CDB, were measured with the molybdenum blue colourimetric method (Murphy and Riley
1958). The P, Mn (assuming to represent Mn oxides; Hermans et al. 2019b) and Si (assuming to
represent metal oxide bound Si; Kostka and Luther III 1994; Rao et al. 2016) in CDB extracts was
determined using ICP-OES.

**2.6. Elemental Mapping of Fe, Mn, P and Ca**

On day 47, an undisturbed core (first 7 cm of surface sediment) was sampled for epoxy resin

embedding for high-resolution elemental mapping (Jilbert et al. 2008; Jilbert and Slomp 2013). The
epoxy-embedded core was split vertically using a rock saw. The surface was smoothed by applying a
0.3 µm alumina powder layer. Elemental maps of Fe, Mn, P and Ca (30 µm resolution) were retrieved
using a Desktop EDAX Orbis µXRF analyser (Rh tube set at 30 kV, 500 µA, 300 ms dwell-time,
equipped with a poly-capillary lens). Similar µXRF maps for Fe, Mn and P in epoxy embedded





surface sediment were obtained for two field sites: (1) the Gulf of Finland, for sediments collected in
June 2016 as described by Hermans et al. (Submitted), and Lake Grevelingen, for sediments collected
in January and May 2012 as described in Sulu-Gambari et al. (2016a; 2018).

**2.7. Fluorescence *In-situ* Hybridisation**

Fluorescence *in-situ* hybridisation (FISH; Pernthaler et al. 2001) was used to microscopically
quantify the abundance of cable bacteria filaments, as described in Seitaj et al. (2015). FISH analysis
was performed on one intact sediment core retrieved at our sampling site, and the sediment cores from
our incubation experiment used for pore water collection at three time points (days 5, 26 and 207).
These cores were sectioned at 0.5 cm depth resolution for the first 2.5 cm. Each sediment slice was
homogenised and fixed with 0.5 mL ethanol (≥99.8% purity), and stored in a 2 mL Eppendorf tube at
-20 °C. For FISH analysis, a volume of 100 µL was retrieved from the Eppendorf tubes and mixed
with a 1:1 solution of PBS/ethanol (500 µL). Then 10 µL of this mixture was filtered through a
polycarbonate membrane (type GTTP; pore size 0.2 µm, Millipore, USA). Cable bacteria were
classified with a *Desulfobulbaceae*-specific oligonucleotide probe (DSB706; 5-ACC CGT ATT CCT
CCC GAT-3') after counter staining with DAPI (1 µg/mL) under an epifluorescence microscope
(Zeiss Axioplan, Germany) at 100x magnification. The abundance of cable bacteria was quantified by
determining the length and diameter of all observed filaments in a field ($105 \times 141$ µm) on the filter at
100x magnification (200 fields per sample). Cable bacterial abundances are expressed as filament
length per volumetric unit (m cm$^{-3}$) or depth integrated per unit area of sediment surface (m cm$^{-2}$),
consistent with previous studies (Schauer et al. 2014; Malkin et al. 2017).

**2.8. Scanning Electron Microscopy**

Cable bacteria filaments were taken from surface sediments from the oxic zone (upper 2 mm)
after 40 days using a microscope and were transferred to a 15 mL centrifuge tube. The tube was filled
to a volume of ~10 mL using ultra clean water, and was subsequently centrifuged at 2100 rpm for 2
min, after which the water was discarded. This washing step was repeated three times. The washed
samples were then transferred to a sample stub, where the sediment was air-dried over-night prior to
gold coating. The filaments were subsequently subjected to scanning electron microscopy (SEM)





imaging on a Phenom ProX Desktop SEM (Phenom-World B.V., the Netherlands) to obtain high-
resolution images, as described in Geerlings et al. (2019). SEM images were generated under 0.1-0.3
mbar vacuum, and a high accelerating voltage (10 or 15 kV).

**2.9. Data Analysis and Calculations**

The diffusive uptake of $O_2$ was calculated from the high-resolution $O_2$ depth profiles using the
PROFILE software package (Berg et al. 1998). Total $H_2S$ ($\Sigma H_2S = H_2S + HS^- + S^{2-}$) was calculated as
a function of the recorded $H_2S$ and pH values, accounting for temperature and salinity (Millero et al.
1988; Jeroschewski et al. 1996).
The EP depth profiles were normalised by subtracting the background EP signal in the
overlying water from the EP depth profiles, to calculate the EP value relative to that in the overlying
water (Damgaard et al. 2014). The electric field in the sediment was calculated from the linear slope
of the EP depth profiles (average of triplicates) in the surface sediments (Risgaard-Petersen et al.
2014). The magnitude of the current density was subsequently calculated from the gradient in the EP,
the so-called electric field, using Ohm's law:

$$J = \sigma_{sed} \cdot E \tag{1}$$

where $J$ represents the magnitude of the current density (mA m$^{-2}$), $\sigma_{pw}$ is the conductivity of
the sediment matrix (S m$^{-1}$) and $E$ (mV m$^{-1}$) represents the electric field. The conductivity of the pore
water was corrected for tortuosity and calculated as a function of the temperature and salinity using
the equations provided by Fofonoff and Millard Jr (1983).
The solute fluxes were calculated as described in Glud (2008)and Rao et al. (2016):

$$J = \frac{\Delta C_{ow}}{\Delta t} \cdot \frac{V_{ow}}{A} \tag{2}$$

where $J$ represents the diffusive flux (mmol m$^{-2}$ d$^{-1}$), $\Delta C_{ow}$ represents the concentration change in the
overlying water (mmol m$^{-3}$), $\Delta t$ is the incubation time (d), $V_{ow}$ is the volume in the overlying water
(m$^3$) and A the surface area of sediment in the core (m$^2$). In our experimental setup, only those fluxes
were measurable for $NH_4^+$, $Fe^{2+}$, $Mn^{2+}$ and $HPO_4^{2-}$, that were >0.08, >0.06, >0.01 and >0.55 mmol m$^{-2}$




d$^{-1}$, respectively. However, for these four solutes, fluxes were always too low to be detected. Hence,
only $Ca^{2+}$ and $H_4SiO_4$ fluxes are presented.

Diffusive downward fluxes of $SO_4^{2-}$ and diffusive upward fluxes of $NH_4^+$, $Fe^{2+}$, $Mn^{2+}$ and

$Ca^{2+}$ were calculated from linearized pore water gradients using Fick's first law (Berner 1980):

$$J = -\phi D_s \cdot \frac{dC}{dz} \tag{3}$$


The molecular diffusion coefficient was calculated as a function of pressure, salinity and

temperature using the R package *marelac* (Soetaert et al. 2010) and corrected for the ambient
tortuosity using the relations listed in Boudreau (1997).
**3. RESULTS**
**3.1. Abundance of Cable Bacteria**

Examination of the top 2.5 cm of the surface sediments using FISH showed the presence of

filamentous cable bacteria (Fig. 1C; Fig. S1). The *in-situ* cable bacterial abundance in the sediment at
our field site was low (14 m cm$^{-2}$). However, after 5 days of incubation in the laboratory, the
abundance increased strongly (724 m cm$^{-2}$). At day 26 the abundance of cable bacteria was even
higher (1035 m cm$^{-2}$). After 207 days, the cable bacterial abundance in the surface sediment was low
again (131 m cm$^{-2}$). SEM imaging confirmed that filaments were indeed cable bacteria (Fig. 1D), as
the external surface of the filament was characterised by a parallel pattern of ridges and grooves along
its latitudinal axis, which is a typical feature of cable bacteria (Cornelissen et al. 2018; Geerlings et al.

2019).

**3.2. High-resolution Depth Profiles of pH, O$_2$ ∑H$_2$S and EP**

High-resolution depth profiles of pH showed the development of a distinct peak near the

sediment-water interface at day 5, and acidification of the pore water in the deeper sediment (Fig.
2A). The width of this pore water acidification zone increased with time and reached its maximum at





day 26, followed by a decrease in the acidification. The distinct pH peak near the sediment-water
interface gradually disappeared after 33 days. The depth of $O_2$ penetration in the sediment remained
constant within the first 40 days of incubation (~1.1 mm) and subsequently moved downwards with
time to 9.6 mm (Fig. 2A; Fig. 3; Fig. S2). The dissolved $\sum H_2S$ concentrations remained low (<5 μM)
throughout the experiment (Fig. 2A). The $\Sigma H_2S$ appearance depth was initially equivalent to the $O_2$
penetration depth, and shifted downwards within 5 days, creating a suboxic zone where $O_2$ and $\Sigma H_2S$
remained below detection (Fig. 2A; Fig. 3). The width of the suboxic zone remained relatively
constant with time (~25 mm; Fig. 3), and only slight decreased after 207 days.

The EP depth profiles indicate a rapid establishment of an electric current after 5 days (0.4

mV; Fig. 2B). The time-series of depth profiles show that the EP increased and also accumulated over
a thicker depth horizon. At day 26 the EP reached its maximum value (1.2 mV), followed by a
decrease with time. Long-distance electron transport was not active in the anoxic control core (Fig.
S3).

**3.3. Diffusive Uptake of $O_2$ and Current Density**

The diffusive $O_2$ uptake of the sediment was highest after 5 days and gradually decreased

with time from ~30 to ~3.6 mmol $m^{-2}$ $d^{-1}$ (Fig. 4A). The current density rapidly increased from day 0
to day 18, from 6 to 128 $e^-$ mmol $m^{-2}$ $d^{-1}$, and then gradually decreased with time (Fig. 4B). The
duplicate measurements show the same trend for the diffusive $O_2$ uptake and the current density,
which indicates that the results are reproducible.

**3.4. Pore Water Profiles**

Concentrations of $NH_4^+$ were low near the sediment-water interface and increased with

sediment depth reaching maximum levels of up to 1.7 mM at depth in the sediment (Fig. 5). The time-
series suggest a gradual decrease in production of dissolved $NH_4^+$ in the sediment leading to
decreasing concentrations with time. The pore water depth profiles of dissolved $SO_4^{2-}$ show a decline
with sediment depth at all time points. However, $SO_4^{2-}$ concentrations remained relatively constant
within the top 2 cm of surface sediment between day 12 and 33. Dissolved $Fe^{2+}$, $Mn^{2+}$ and $Ca^{2+}$ all





show the development of distinct peaks in the pore water with time, and after 40 days those peaks
disappear again. Pore water concentrations of $HPO_4^{2-}$ generally increased with sediment depth for all
time points, and concentrations within the top 2 cm were below the detection limit indicating removal.
Dissolved $H_4SiO_4$ increased with sediment depth reaching concentration of up to 1 mM.

**3.5. Diffusive Fluxes**

Calculated diffusive fluxes of $NH_4^+$ into the oxic zone decreased during the incubation
experiment from 4.7 to 1.8 mmol m$^{-2}$ d$^{-1}$ (Fig. 6A; Fig. S4; Table S3). Rates of $SO_4^{2-}$ reduction
estimated from the linear gradient of the decrease in pore water $SO_4^{2-}$ in the surface sediment with
depth generally also showed a decrease with time (Fig. 6B; Fig. S5; Table S3). The upward diffusive
flux of $Fe^{2+}$ greatly increased from day 5 to day 12 and then gradually decreased with time (Fig 6C;
Fig. S6; Table S3). The upward diffusive flux of Mn showed an increase in the initial stage of the
experiment and reached its maximum at day 18, followed by a decrease with time (Fig. 6D; Fig. S7;
Table 3). The upward diffusive flux of $Ca^{2+}$ showed no clear trend with time, however after 207 days
the flux became extremely low (Fig. 6E; Fig. S8). The upward diffusive flux of $H_4SiO_4$ also showed
an increase in the initial stage of the experiment, and reached its maximum at day 12, followed by a
decrease with time (Fig. 6F; Fig. S9).

**3.6. Solid-phase Profiles**

The surface sediment in the oxic cores became more enriched in Fe oxides with time, with
concentrations increasing from 53 to 485 μmol g$^{-1}$ (Figure 7). The deeper sediment in the oxic cores
and the entire anoxic control core had low or no Fe oxides. At day 5, FeS was strongly depleted
within the top 1 cm of the surface sediment and was gradually lost further with time. At day 621, most
of the FeS within the top 2.5 cm of the surface sediment had been dissolved. The anoxic core did not
show such a depletion of FeS in the surface sediment and even showed a slight increase in FeS. Solid-
phase siderite remained rather constant with depth from day 5 to 33, but afterwards was gradually lost
from the surface sediment. At day 621 a large proportion of the siderite was dissolved within the top 2
cm. Solid-phase siderite concentrations remained constant with depth in the anoxic control core.



Solid-phase depth profiles of Mn oxides, metal bound P and metal oxide bound Si all showed a
gradual increase in the surface sediment with time.

**3.7. High-resolution Elemental Mapping**

High-resolution desktop μXRF mapping of Fe, Mn, P and Ca of our core after 47 days of
incubations revealed a subsurface sediment layer highly enriched in Fe and P (Fig. 8A). Subsurface
enrichments in Fe, P and Mn in relatively thin layers were also observed in sediments populated by
cable bacteria in the Gulf of Finland and Lake Grevelingen (Fig. 8B and C). In the latter system, the
layers enriched in Fe, P, Mn broadened upon recolonization by macrofauna (Fig. 8D).

## 3. DISCUSSION

**4.1. Metabolic Activity of Cable Bacteria**

Cable bacteria in our incubation experiment demonstrated a rapid growth, since their
abundance greatly increased after 5 days, and reached its peak at day 26 (Fig 1C). Such high
abundances are similar to those observed in previous experiments, in which FeS-rich marine
sediments from Aarhus Bay and Lake Grevelingen were incubated (Schauer et al. 2014; Burdorf et al.
2018). The activity of cable bacteria exerted a strong impact on the pore water depth profiles of pH,
$O_2$, and $\sum H_2S$, as evident from the development of a pH maximum near the sediment-water interface,
the strong pore water acidification in the deeper sediment and the development of a suboxic zone (Fig.
2A). These pore water depth profiles resemble the distinct biogeochemical fingerprint typical for
active cable bacteria, as observed in previous laboratory incubation experiments (Risgaard-Petersen et
al. 2012; Malkin et al. 2014; Schauer et al. 2014; Vasquez-Cardenas et al. 2015; Rao et al. 2016;
Burdorf et al. 2018). The widening of the suboxic zone with time (Fig 3) is a consequence of the
downward expansion of the cable bacteria filament network (Schauer et al. 2014; Vasquez-Cardenas
et al. 2015).
The EP depth profiles demonstrated that long-distance electron transport by cable bacteria
was already active 5 days after the start of the experiment, as indicated by the increase of EP at depth
to 0.4 mV). With time, the EP signal increased to higher values and also accumulated over a thicker



depth horizon (Fig. 2B), indicating that cable bacteria activity both increased and extended to deeper
sediment depth, which is also a consequence of the downward expansion of cable bacteria filaments.
The EP reached a maximum after 18 days (1.3 mV; Fig. 2B) in concert with the highest current
density of ~130 mmol e$^-$ m$^{-2}$ d$^{-1}$ (Fig 4B). This maximum EP value and current density are similar in
magnitude to those found in sediment incubations with seawater with a similar salinity (Damgaard et
al. 2014). From day 18 onwards the EP and current density flux gradually decreased with time to 13
mmol e$^-$ m$^{-2}$ d$^{-1}$ after 207 days (Fig. 4B), which implies a decrease in the metabolic activity of cable
bacteria. The suboxic zone persisted long after the current density had decreased (Fig. 3).

To summarise, the metabolic activity of cable bacteria was likely highest between day 18 and

day 26 based on the cable bacterial abundances, the extent of acidification of the pore water and the
current density (Fig 1C; Fig 2A and Fig 4B).

**4.2. Organic Matter Degradation**

Ammonium fluxes are assumed to reflect rates of anaerobic degradation of organic matter (Fig.

5), and the observed decline during the experiment coincides with the decrease in activity of cable
bacteria based on the EP profiles and current density (Fig. 2B; Fig 4B). This suggests that the
availability of easily degradable organic matter plays a role in sustaining the metabolic activity of
cable bacteria, most likely by controlling the rate of $SO_4^{2-}$ reduction (Nielsen and Risgaard-Petersen

2015).

Rates of $SO_4^{2-}$ reduction estimated from the linear gradient of the decrease in pore water $SO_4^{2-}$ in

the surface sediment with depth indeed also showed a decline during the experiment. We note,
however, that a direct measurement of $SO_4^{2-}$ reduction rates (Fossing and Jørgensen 1989; Kallmeyer
et al. 2004) would provide a better indicator, because $SO_4^{2-}$ estimated from pore water profiles are in
general lower than rates estimated from tracer experiments (Hermans et al. 2019a; Sandfeld et al.
2020). Another cause for a slight underestimation of our $SO_4^{2-}$ reduction rates, is due to the effect of
the electric field imposed by cable bacteria, which is not taken into account in Fick's law.





The metabolic activity of cable bacteria can lead to the production of $SO_4^{2-}$ in the suboxic zone
via anodic sulphide oxidation (Risgaard-Petersen et al. 2012; Rao et al. 2016). We suspect that this
also explains the lack of change in pore water $SO_4^{2-}$ with depth in the upper 2 cm of the sediment in
our experiment between 12 and 40 days (Fig. 5). Despite relatively high $SO_4^{2-}$ reduction rates ranging
from 5.4 to 17.6 mmol $m^{-2}$ $d^{-1}$ (Fig. 6B; Table S3), pore water concentrations of $\Sigma H_2S$ remained very
low throughout the experiment (Fig. 2A). This is likely due to the direct consumption of $\Sigma H_2S$
through the activity of cable bacteria, preventing $\Sigma H_2S$ from accumulating in the pore water, or
alternatively, precipitation of FeS by dissolved $Fe^{2+}$ released from the dissolution of siderite.
Laboratory experiments have shown that S-oxidation by cable bacteria can play a dominant role
in the $O_2$ uptake of coastal sediments (Nielsen et al. 2010; Schauer et al. 2014; Nielsen and Risgaard-
Petersen 2015), and model analysis predicts up to 93% of the total $O_2$ uptake (Meysman et al. 2015).
When we plot diffusive uptake of $O_2$ against the current density (i.e. upward flux of electrons towards
the oxic zone), a linear relationship – with some scatter - emerges for days 12 to 621 (Fig. 9).
However, the data points for day 0 and 5 during the initial stages of our experiment do not follow this
linear relationship. We explain these findings as follows: At day 0, the cable bacteria were not active
yet and other processes, such as aerobic respiration and oxidation of $NH_4^+$ and other solutes (Table 2)
and solids (FeS) dominated the consumption of $O_2$. At day 5 and 12, the activity of cable bacteria and
the oxidation of reduced products from anaerobic degradation of organic matter both contributed to
consumption of $O_2$. From day 12 onwards, both the $O_2$ consumption and electron flux follow a
downward decrease with time (Fig. 9). If cable bacteria would account for all of the $O_2$ consumption,
a ratio between the diffusive uptake of $O_2$ and the current density of 1:4 is expected (Fig. 1A; Nielsen
et al. 2010). We find that from day 12 onwards, most data points plot rather close to the line for this
1:4 relationship (Fig. 9), suggesting that cathodic $O_2$ reduction by cable bacteria is responsible for
nearly all $O_2$ consumption in the sediment (in line with the model results of Meysman et al. 2015).
This however poses a problem for the nitrogen budget, because our data indicate complete removal of
the $NH_4^+$ that diffuses upward into the oxic zone (Fig. 6A), and based on the solute fluxes, no escape
to the overlying water (see section 2.4). This implies substantial $O_2$ consumption due to nitrification



(Table 2). These findings can be explained, however, if we assume that the $NO_3^-$ that is being formed
near the sediment-water interface is also used for the metabolic activity of cable bacteria. It has been
shown that cable bacteria can couple the oxidation of $\sum H_2S$ to $NO_3^-$ in the absence of $O_2$ (Marzocchi
et al. 2014). Our data suggest that this process may also occur in sediments where $O_2$ is present in
concert with $NO_3^-$ near the sediment-water interface. Another explanation is that cable bacteria might
consume $O_2$ directly above the sediment-water interface, as recently has been proposed by Burdorf et
al. (2018). Lastly, the current density might be slightly overestimated, since it ignores other sources
that can create an electric potential, such as the diffusion potential (Revil et al. 2012; Nielsen and
Risgaard-Petersen 2015).

**4.3. Impact of Cable Bacteria on Fe, Mn and S Cycling**

The activity of cable bacteria had a strong impact on the biogeochemistry of the surface sediment
in our experiment (Fig. 7). Cable bacteria activity induced an intense acidification of the pore water in
the suboxic zone (Fig 2A), which led to the dissolution of Fe and Mn minerals in deeper sediment
layers, as can be inferred from the sharp maxima in dissolved $Fe^{2+}$ and $Mn^{2+}$ in the pore water
reaching concentrations of up to ~1700 and ~80 μM, respectively (Fig. 5). The twenty-fold higher
dissolved $Fe^{2+}$ concentrations with respect to pore water $Mn^{2+}$ can be attributed to the relatively higher
availability of FeS and siderite compared to the availability of Mn carbonates in the sediment that was
used for incubation (Lenstra et al. 2020). The peaks in dissolved $Fe^{2+}$ and $Mn^{2+}$ in the pore water
broadened over time spanning a depth of >5cm (Fig. 5; Fig. S6; Fig. S7).
The upward diffusive flux of dissolved $Fe^{2+}$ and $Mn^{2+}$ was highest after 12 days, reaching values
of up to 3.16 and 0.16 mmol m$^{-2}$ d$^{-1}$ respectively. Fluxes subsequently gradually decreased with time
(Fig. 6C and D). The continuous upward diffusion of dissolved $Fe^{2+}$ and $Mn^{2+}$ led to enrichments of
poorly crystalline Fe and Mn oxides in the surface sediment (Fig. 7). Despite high upward fluxes of
dissolved $Fe^{2+}$ and $Mn^{2+}$ towards the sediment-water interface, our solute flux incubations indicate
there was little escape of $Fe^{2+}$ and $Mn^{2+}$ to the overlying water (see section 2.4). This implies that all
$Fe^{2+}$ and $Mn^{2+}$ that diffused upward was precipitated as Fe and Mn oxides upon contact with $O_2$ or
$NO_3^-$ (Buresh and Moraghan 1976; Kuz'minskii et al. 1994; Straub et al. 1996). Little or no escape of





dissolved $Fe^{2+}$ from the sediment into the overlying water, was suggested previously for a field site
with active cable bacteria based on diffusive flux calculations (Lake Grevelingen; Sulu-Gambari et al.
2016a) and was determined in flux incubations of cores during a laboratory experiment with cable
bacteria (Rao et al. 2016).
At the start of the experiment, the sedimentary FeS content was (~25 µmol $g^{-1}$), which is not
unusual for coastal sediments on the north-western Black Sea margin (Wijsman et al. 2001), but is
low compared to sediments in eutrophic coastal systems (e.g. Morgan et al. 2012; Kraal et al. 2013;
Hermans et al. 2019a). The solid-phase depth profiles reveal a gradual removal of the FeS in the
surface sediment in our experiment over time (Fig. 7). At the end of our experiment (621 days), there
was no longer any FeS within the top 1.5 cm of the sediment. While approximately 90 mmol $m^{-2}$ of
FeS was removed from the surface sediment within the first 5 days, a total of ~240 mmol $m^{-2}$ was
removed after 621 days (Fig. 10; Table 3). Likely, part of the FeS that was removed from the surface
sediment within the first 5 days was removed through oxidation upon contact with $O_2$, rather than the
metabolic activity of cable bacteria itself. The pore water acidification associated with cable bacteria
activity led to a strong loss of siderite within the top 2 cm of the sediment, with a total removal of
~560 mmol $m^{-2}$ during the experiment (Fig. 7; Fig. 10; Table 3). The depletion of sedimentary FeS
and siderite was directly proportional to the formation of Fe oxides near the sediment-water interface
(Fig. 10), and accounted for 30% and 70% of the Fe oxides, respectively (Table 3).
With these data we cannot accurately determine the role of FeS versus $SO_4^{2-}$ reduction in
supplying the $\sum H_2S$ sustaining the activity of cable bacteria throughout the experiment. This is
primarily related to the variability between cores, and for this type of calculation, the low temporal
resolution of sampling. However, we can make an estimation of the relative role of $SO_4^{2-}$ reduction
and FeS dissolution in $\sum H_2S$ production, based on the pore water profiles of $SO_4^{2-}$ and dissolved $Fe^{2+}$,
and the solid-phase mass balance of FeS and siderite (Fig. 6B and C; Table 4).  This estimation shows
that $SO_4^{2-}$ was mainly responsible for $\sum H_2S$ production, accounting for 85-99% (Table 4), and thus
that the dissolution of FeS only played a minor role in providing $\sum H_2S$.



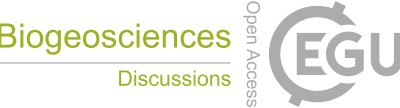

### 4.4. Impact of Cable Bacteria on Ca, P and Si Cycling

Cable bacteria activity is known to lead to dissolution of Ca carbonates, because of the strong acidification of the pore water (Risgaard-Petersen et al. 2012; Rao et al. 2016). We indeed find similar maxima in pore water $Ca^{2+}$ during the experiment (Fig. 5) and a high upward flux of $Ca^{2+}$ (up to ~18 mmol $m^{-2}$ $d^{-1}$; Fig. 6E; Fig. S8) of which a substantial fraction (up to ~55%) escapes to the overlying water (Fig. S10; Table S4), which is consistent with a previous incubation experiment Rao et al. (2016).

Pore water depth profiles of $HPO_4^{2-}$ reveal a production at depth and removal of all upward diffusing $HPO_4^{2-}$ within the first 1-3 cm of the surface sediment (Fig. 5). A major proportion of this $HPO_4^{2-}$ is bound to Fe oxides (Fig. 7). Given that a large proportion of the Fe oxides in our sediment cores derive from the dissolution of siderite, this suggests that the buffer mechanism that delays the benthic release of $HPO_4^{2-}$ through retention of P associated with newly formed Fe oxides (Sulu-Gambari et al. 2016b), might also be active in systems that are relatively poor in sedimentary FeS.

The shape of the pore water $HPO_4^{2-}$ profiles suggests that some of the $HPO_4^{2-}$ is removed below the zone where Fe and Mn oxides are present (Fig. 5; Fig. 7). A possible explanation could be the formation of vivianite, an Fe(II) phosphate mineral. Vivianite formation in sediments typically occurs when pore water levels of $Fe^{2+}$ and $HPO_4^{2-}$ are high and concentrations of $\Sigma H_2S$ are low (Nriagu 1972), as observed in our study. In our experiment, free $\Sigma H_2S$ does not accumulate in the pore water, which we attribute to removal through the activity of cable bacteria and FeS formation at depth (Fig. 2A; Fig. 7). Hence, cable bacteria may create a geochemical niche that allows the formation of vivianite in the suboxic zone. Further work with sediments with higher P concentrations would be needed to assess this with direct measurement techniques, such as X-ray spectroscopy (Egger et al. 2015; Kraal et al. 2017; Sulu-Gambari et al. 2018). Other sediment P pools, i.e. organic, authigenic and detrital P remained constant over time, indicating that the P contents determined for discrete sediment slices using sequential extractions were not affected by pore water acidification as a result of cable bacteria activity (Table S5).



Pore water $H_4SiO_4$ profiles show a typical increase with depth as observed upon dissolution of
biogenic silica in marine sediments (Aller 2014). Fluxes of $H_4SiO_4$ towards the sediment-water
interface range up to ~2.8 mmol m$^{-2}$ d$^{-1}$ and gradually decreased with time (Fig. 6F; Fig. S9). The
results of the solute flux incubations indicate that most of this $H_4SiO_4$ escaped to the overlying water
(ranging from 28 to 92%; Table S4; Fig. S10). The decline in the benthic release flux of $H_4SiO_4$
contrasts with results of a previous incubation experiment by Rao et al. (2016) with similar pore water
concentrations of $H_4SiO_4$ reaching values up to ~1 mM. In their study, the flux remained constant
over time, possibly because of differences in the amount of biogenic Si in the sediment. The solid-
phase metal oxide bound Si pool in the surface sediment increased directly proportional to the
formation of Fe oxides throughout the experiment (Fig. 7). Silica is known to absorb to Fe oxides
(Sigg and Stumm 1981; Davis et al. 2002). Hence, the results suggest that the Fe oxides formed
through the activity of cable bacteria captured some of the upward diffusing $H_4SiO_4$.

**4.5. Sediment Marker for Cable Bacteria Activity**

Visual observations of core photographs reveal the gradual development of an orange layer (oxic
zone) up to 9 mm thick, overlying a grey layer (suboxic zone) and a black layer (sulphidic zone)
during the experiment (Fig. S11). This colour zonation is typical for sediments that have been
geochemically affected by cable bacteria activity (Nielsen and Risgaard-Petersen 2015; Sulu-Gambari
et al. 2016a). High-resolution elemental maps of our sediments reveal the development of a ~0.3 mm
thin subsurface layer highly enriched in Fe oxides and associated P, 47 days after the start of the
incubation (Fig. 8A). Below, we describe the formation of this layer in more detail and explain why
such subsurface enrichments, detected with µXRF, may act as an additional sediment marker for
present or recent cable bacteria activity, also in cases where visual observations are not conclusive.
During the experiment, $O_2$ penetration varied within a narrow range and was initially fixed
between 1 and 2 mm depth (Fig. 3A), with the layer highly enriched in Fe forming mostly at a depth
of 2 mm (Fig. 8A). This can be explained by rapid oxidation of upward diffusing $Fe^{2+}$ upon contact
with $O_2$ (and possibly $NO_3^-$; Fig. 6C). Directly, above the Fe oxide layer a broader ~0.8 mm thick Mn
oxide layer was observed (Fig. 8A). This contrast in zonation between Fe and Mn is likely due to the



slower oxidation kinetics of $Mn^{2+}$ compared to $Fe^{2+}$ (Burdige 1993; Luther 2010; Learman et al.

2011).

While the Fe oxide layer is clearly enriched in P, we also observed a second layer enriched in P

close to the sediment-water interface (Fig. 8A). In this layer, P is strongly correlated with Ca. This
layer likely consists of carbonate fluorapatite (CFA), a Ca-P mineral, which is typically formed in
marine sediments (Van Cappellen and Berner 1988; Ruttenberg and Berner 1993). Possibly, the high
pore water pH near the sediment-water interface (resulting from cathodic $O_2$ reduction by cable
bacteria; Fig. 2A), promotes apatite formation (Bellier et al. 2006), and the elevated $Ca^{2+}$
concentrations (Fig. 5) created a biogeochemical niche for the formation of CFA.

Such focusing of Fe, Mn, P and associated elements within a thin subsurface layer, as a

consequence of cable bacteria activity, also occurs in the field. This was demonstrated by Hermans et
al. (Submitted) in a study of a coastal site in the Gulf of Finland where cable bacteria were recently
active. Here, µXRF mapping of resin embedded sediments revealed strong focusing of Fe oxides,
Mn(II) phosphates and Fe bound P within a 3 mm thick layer near the sediment-water interface (Fig.
8B). A re-assessment of the µXRF data of Sulu-Gambari et al. (2016a; 2018) of surface sediments
with active cable bacteria from seasonally hypoxic marine Lake Grevelingen in January also revealed
similar subsurface enrichments in Fe, Mn and P (Fig. 8C). Importantly, no visual signals for cable
bacteria based on the colour pattern of the sediment were observed at the time.

Macrofaunal activity likely counteracts or prevents strong focusing of Fe oxides and associated P

within such a thin subsurface layer at field sites. Bioturbation, i.e. mixing of the sediment, typically
leads to oxidation from the sediment surface downwards (Norkko et al. 2012). Bioirrigation can
efficiently pump $O_2$ into the pore water and thereby enhance the oxidation of dissolved $Fe^{2+}$
(Kristensen et al. 2012; Norkko et al. 2012), but is not expected to lead to such a sharp oxidation front
(Norkko et al. 2012; Hermans et al. 2019a). This is also evident from high-resolution elemental maps
of the surface sediment from Lake Grevelingen in May, which shows the disappearance of the thin



layer highly enriched in Fe and P formed by cable bacteria in January as a consequence of
macrofaunal activity in May (Fig. 8D; Seitaj et al. 2015; Sulu-Gambari et al. 2016b).
We conclude that the focusing of Fe, Mn and associated P within a thin layer below the sediment-
water interface is likely a consistent feature in sediments populated by active cable bacteria and may
be a marker for their recent activity. When using standard techniques for sediment sampling (i.e. core
slicing and chemical analysis of these slices), these layers may be missed due to the relatively coarse
depth resolution. Hence, µXRF mapping of epoxy embedded sediment is recommended.
**4.6. Cable Bacteria Activity at the Field Site**
We can only speculate about the possible *in-situ* relevance of cable bacteria at the coastal site
in the western Black Sea where the sediment for our incubation was collected. At this site, both
bivalves (up to ~7200 ind. m$^{-2}$) and polychaetes (up to ~1700 ind. m$^{-2}$) were observed at the time of
sampling (Lenstra et al. 2019). Macrofauna can inhibit the activity of cable bacteria through
bioturbation by physically cutting and damaging the filaments, rendering them unable to transport
electrons (Malkin et al. 2014). Recent work has shown, however, that in some cases, cable bacterial
communities can also thrive in sediments with macrofauna (Burdorf et al. 2017; Malkin et al. 2017;
Aller et al. 2019). In a study of bivalve reefs, cable bacteria were found to efficiently remove highly
toxic $\Sigma H_2S$, which is beneficial for bivalves (Malkin et al. 2017). Cable bacteria can also be abundant
in bioturbated deposits, when associated with stable subdomains of the bioturbated zone, such as
worm tubes (Aller et al. 2019). In such settings, a more complex precipitation pattern, e.g. along tube
linings is observed (Aller et al. 2019), than described here for laboratory experiments with defaunated
sediments and field sediments with an impoverished macrofaunal population (Fig. 8A). Further field
studies are required to assess the role of cable bacteria at our field site, preferably including an
assessment of the burrow structures.
**Conclusions**
The results of our laboratory incubation (with a total duration of 621 days) highlight the
strong impact of cable bacteria on Fe, Mn, P and S dynamics in coastal sediments. The strong acidity



of the pore water associated with the activity of cable bacteria, which was monitored using
microsensor profiling of the EP during the experiment, led to dissolution of FeS and siderite and
formation of Fe and Mn oxides and Ca-P in mineral form near the sediment surface. Both FeS and
$SO_4^{2-}$ reduction provided the $\sum H_2S$ required by cable bacteria to sustain their activity. Pore water
$\sum H_2S$ was always low (<5 µM). Using µXRF mapping of epoxy embedded sediment, we show that
the activity of cable bacteria led to the development of a thin subsurface sediment layer (0.3 mm) that
was highly enriched in Fe and P. The position of this layer in the sediment was directly proportional
to the $O_2$ penetration depth during the experiment. We show that a similar layer highly enriched in Fe
and P was also formed in sediments of field locations populated by cable bacteria (i.e. marine Lake
Grevelingen and the brackish Gulf of Finland). We suggest that such layers, which are not necessarily
visible by eye, may be used as a marker of cable bacteria activity in sediments with low macrofaunal
activity.
**Acknowledgements**

We are grateful to the captain and crew of R/V *Pelagia* for their support during the expedition

(64PE411). We thank S. Hidalgo-Martinez for the FISH analysis and for the SEM image of the cable
bacteria filaments. We thank F. Sulu-Gambari for sharing µXRF data for sediments from Lake
Grevelingen. We also thank N. Geerlings, Z. Wang, K. Wunsch, T. Hakkert, W.K. Lenstra, N.A.G.M.
van Helmond, P. Kraal, M. Egger, A. Tramper, T. Zalm and J.J. Mulder for analytical support. This
research was funded by the Netherlands Organisation for Scientific Research (NWO), Vici Grant
865.13.005 to CPS. Further support was provided by Research Foundation Flanders FWO Grant
G038819N, NWO Vici Grant 016.VICI.170.072 to FJRM and by the Danish National Research
Foundation [Agreement nos. DNRF104 and DNRF136].

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


**FIGURES AND TABLES**

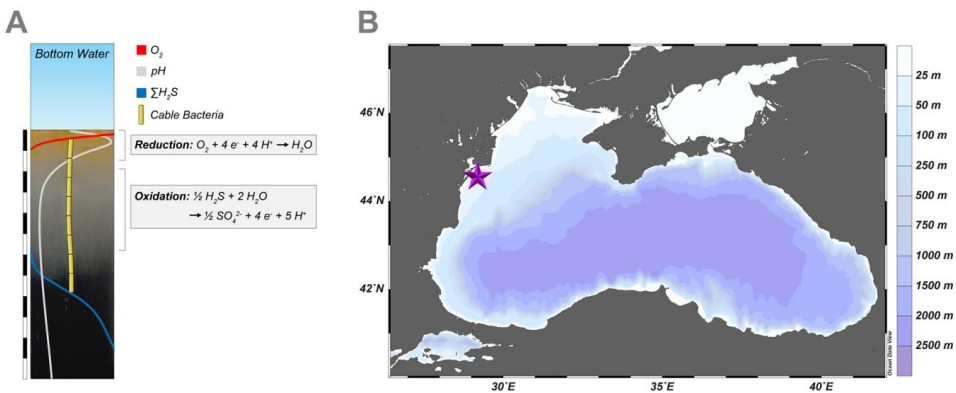

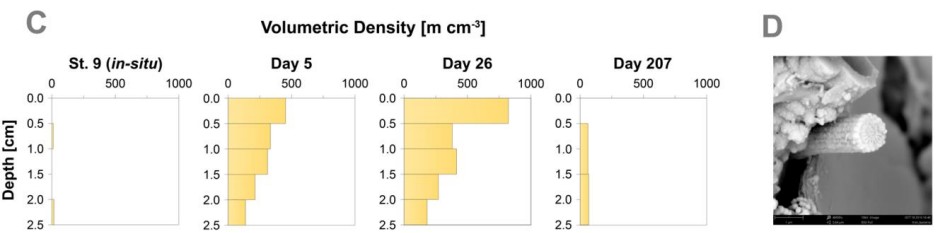


**Fig. 1. (A)** Geochemical pore water fingerprint typical for cable bacteria activity. This fingerprint is defined by a distinct pH
profile (light grey line) and a sub-oxic zone that is devoid of $O_2$ (red line) and $H_2S$ (blue line). The cable bacteria filaments
are depicted in yellow. On the background, the sediment core photograph, taken 278 days after the start of the experiment,
shows a distinct colour zonation where (1) the oxic zone displays an orange colour (2) the suboxic zone has a grey colour
and (3) the sulphidic zone has a black colour. The scale bar denotes a distance of 6 cm, with 0.5 cm intervals. **(B)**
Bathymetric map of the Black Sea. The purple star indicates the location of our study site (44°34.93`N, 29°11.38`E), which
was sampled with R/V *Pelagia* in September 2015. Further details are provided in Lenstra et al. (2019). **(C)** Volumetric
density of cable bacteria [m cm$^{-3}$] in the top 2.5 cm of the sediment, for *in-situ* as well as for three time points during the
incubation experiment **(D)** SEM image of a cable bacteria filament that was extracted from the surface sediment after 40
days.



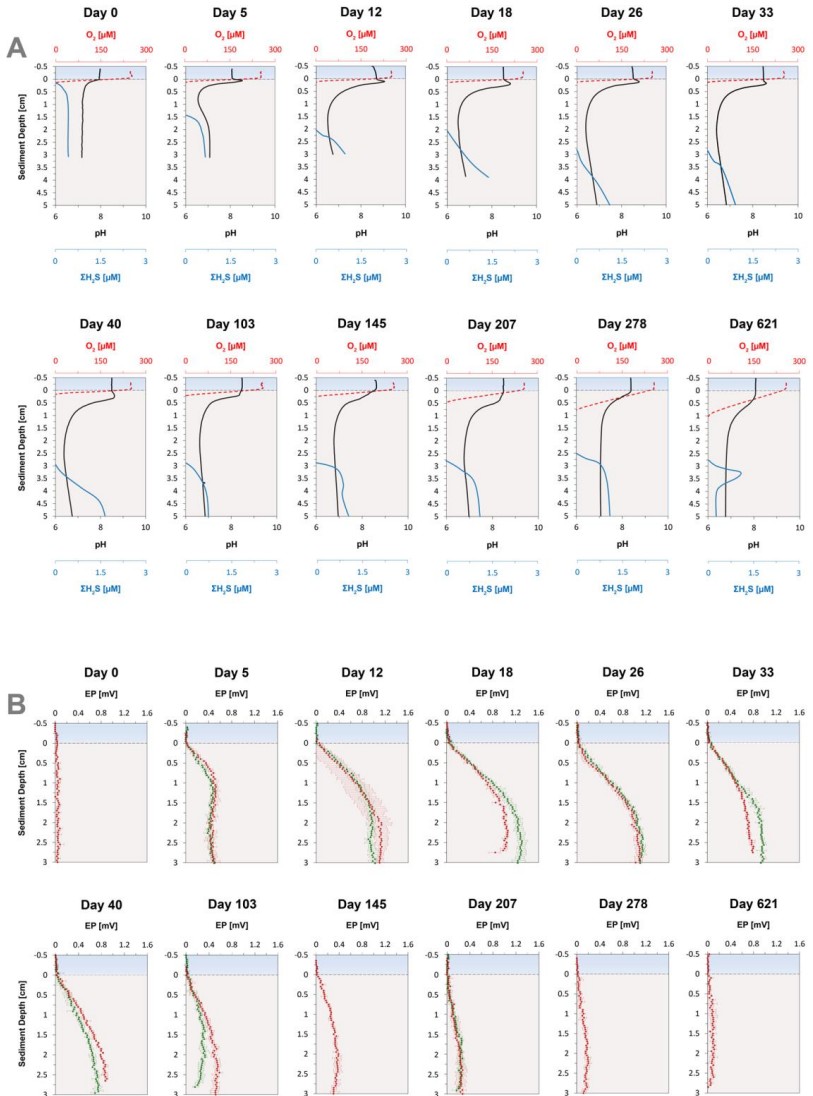

**Fig. 2. (A)** Time-series of the pore water pH (black), $O_2$ (red) and $\sum H_2S$ (blue) signatures of the incubated sediment. **(B)** Development of the EP depth profile in the incubated sediment over time. The dashed-line at 0 mm depth represents the sediment-water interface. The blue boxes indicate the overlying water, whereas the underlying light grey boxes represent the sediment. The EP depth profiles represent an average of 3 replicate measurements. The error bars indicate the minimum and maximum EP values that were observed. The green depth profiles represent duplicate measurement performed on a different core.





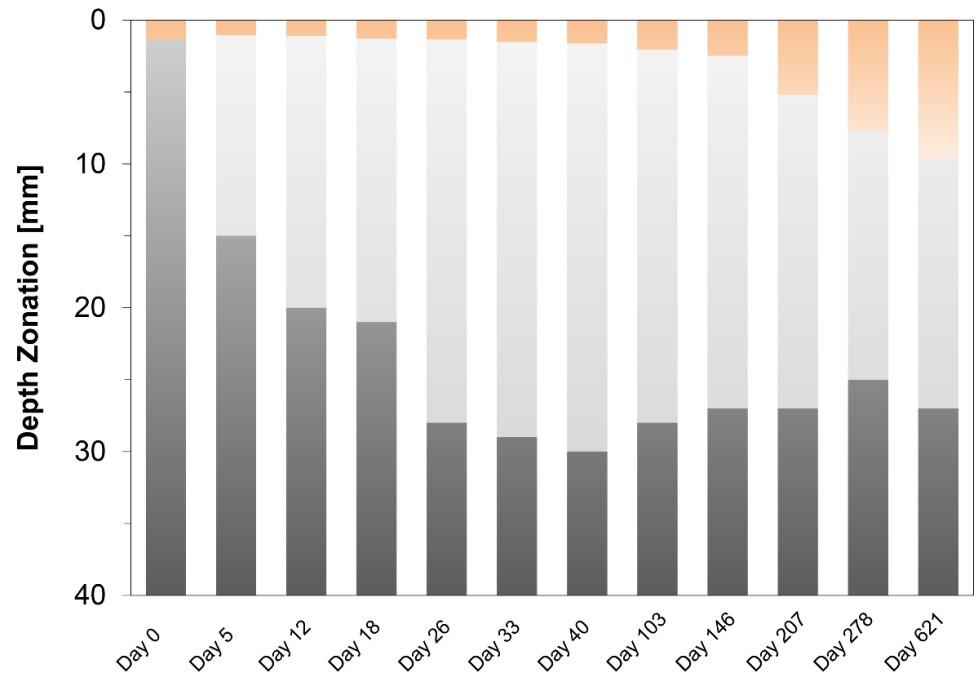


**Fig. 3.** Time-series of the development of the oxic zone (orange), suboxic zone (light grey) and the anoxic/sulphidic zone

(dark grey) in the sediment. These zones were calculated from 3 replicate measurements on two different cores.




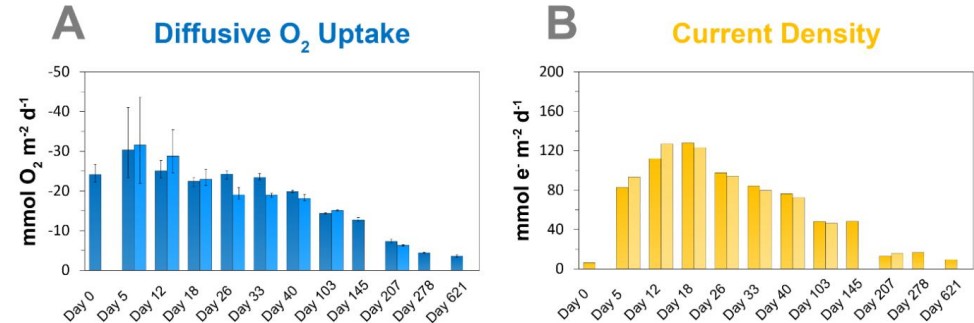


**Fig. 4.** Time-series of the **(A)** diffusive $O_2$ uptake in mmol $O_2$ m$^{-2}$ d$^{-1}$ and **(B)** current density as a consequence of long-distance electron transport (e$^-$) in mmol e$^-$ m$^{-2}$ d$^{-1}$ in the sediment incubation.






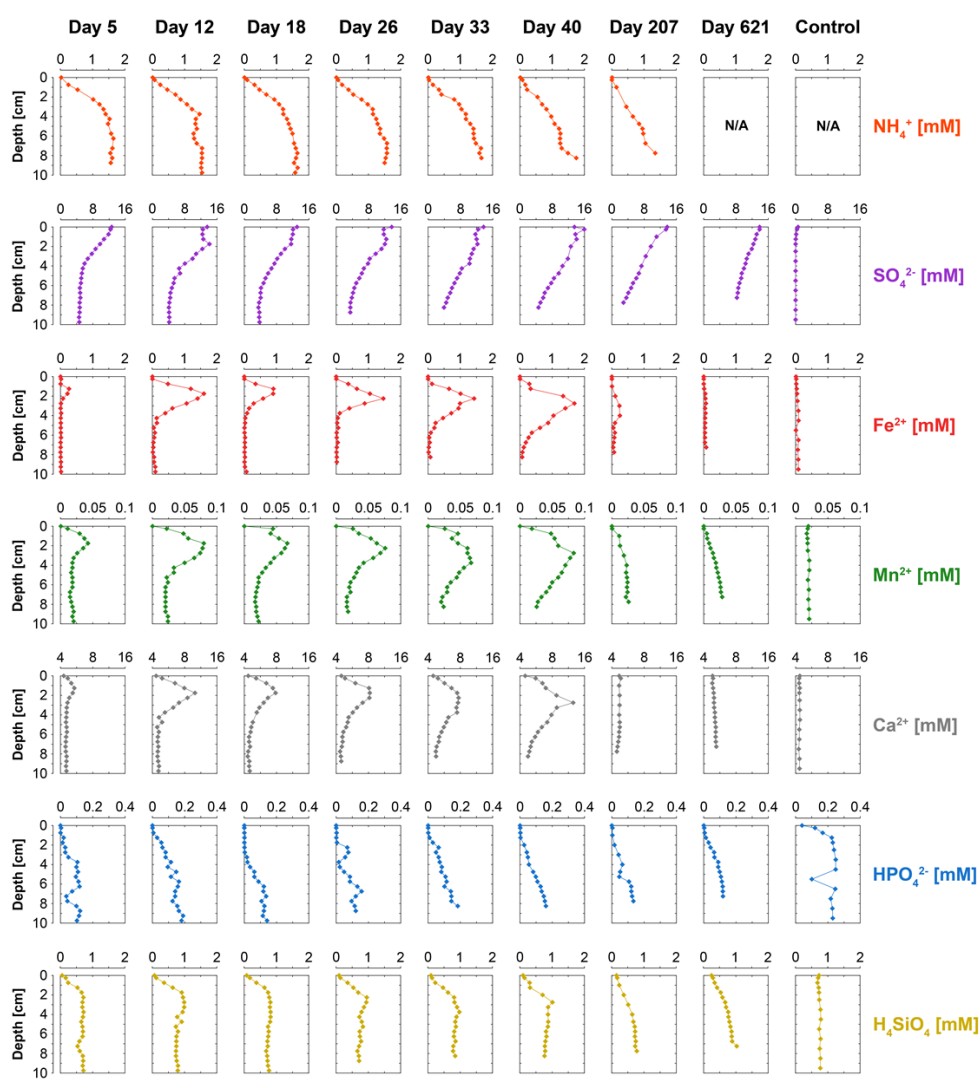

**Fig. 5.** Time-series of pore water depth profiles of $NH_4^+$ (orange), $SO_4^{2-}$ (purple), $Fe^{2+}$ (red), $Mn^{2+}$ (green), $Ca^{2+}$ (grey), $HPO_4^{2-}$ (blue) and $H_4SiO_4$ (yellow). The control core was sampled at day 621.

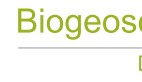
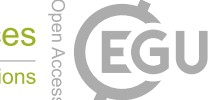

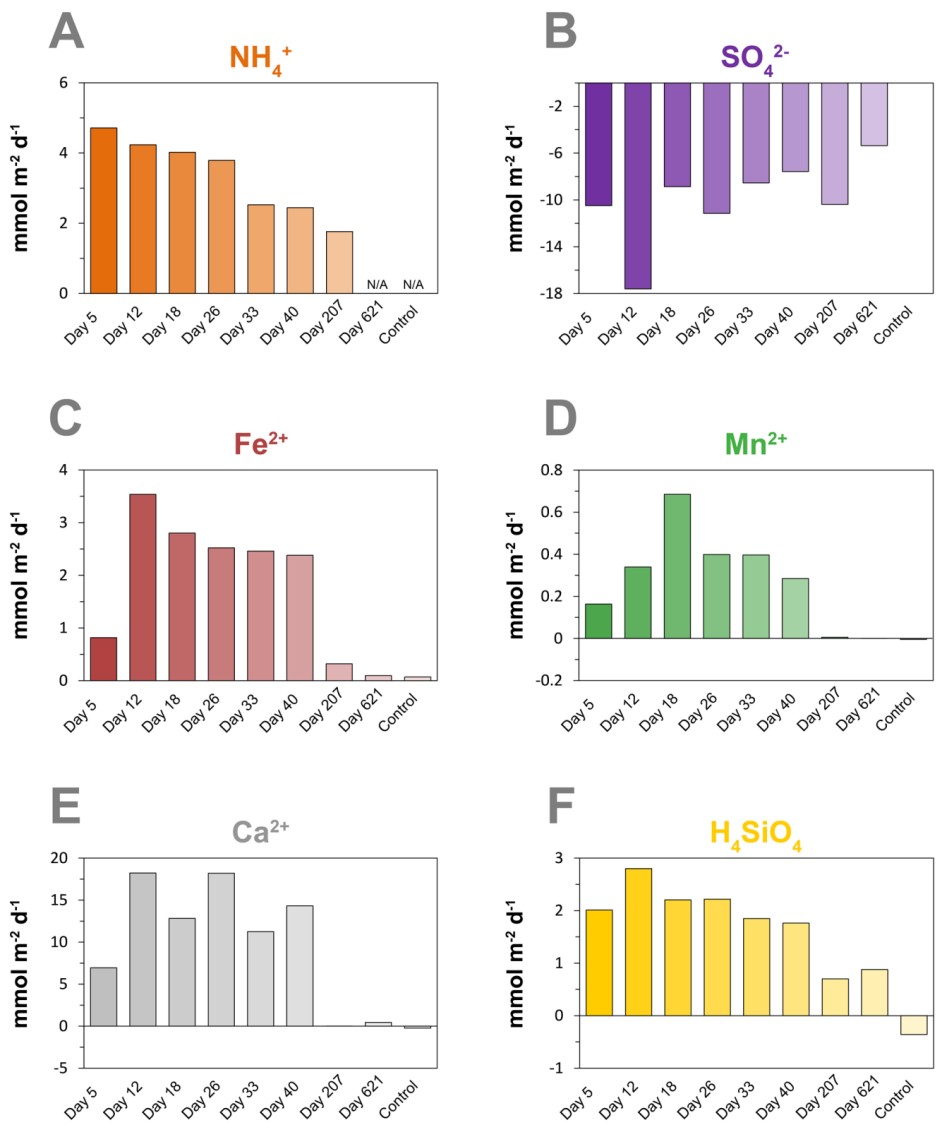

**Fig. 6.** Time-series of diffusive fluxes calculated from the linear gradient of the pore water profiles of **(A)** $NH_4^+$, **(B)** $SO_4^{2-}$, **(C)** $Fe^{2+}$, **(D)** $Mn^{2+}$, **(E)** $Ca^{2+}$ and **(F)** $H_4SiO_4$ in mmol m$^{-2}$ d$^{-1}$ towards the oxic zone of the sediment, based on the linear pore water gradients (Section 1.6; Fig. S3-S4). Here, a positive value indicates an upward flux, whereas a negative value represents a downward flux. N/A = not available. The control core was sampled at day 621.





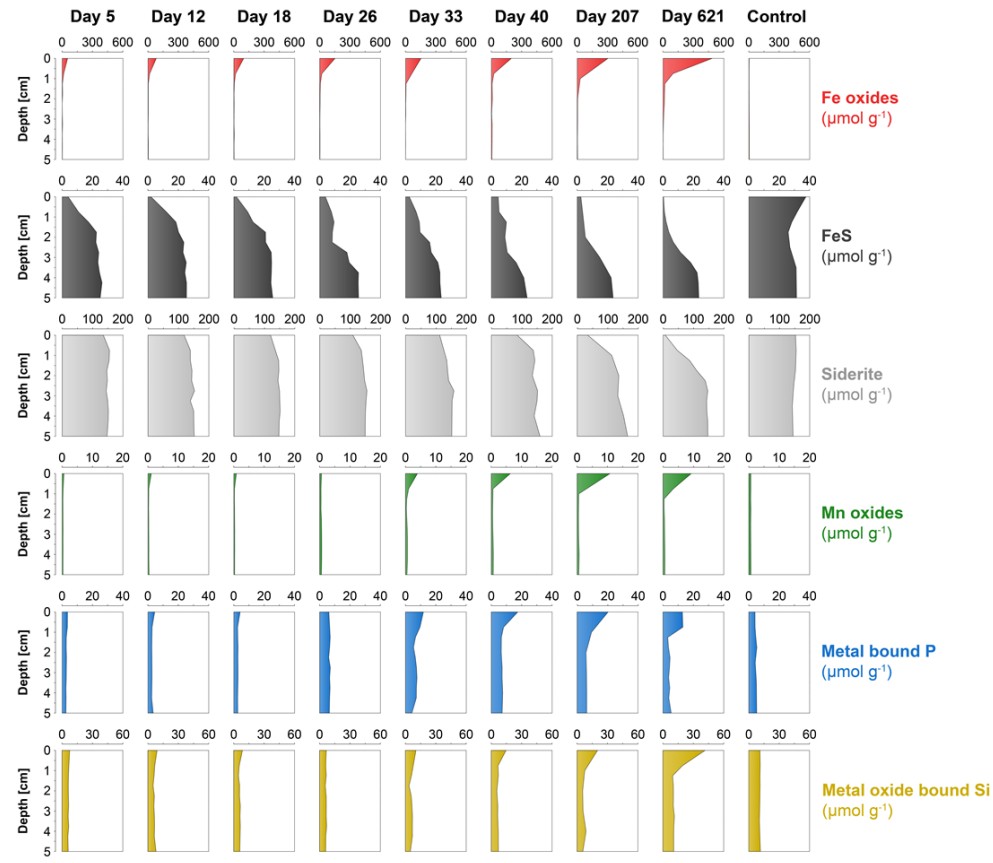

**Fig. 7.** Time-series of solid-phase depth profiles of Fe oxides (red), FeS (black), siderite (grey), Mn oxides (green), metal bound P (blue) and metal oxide bound Si (yellow).
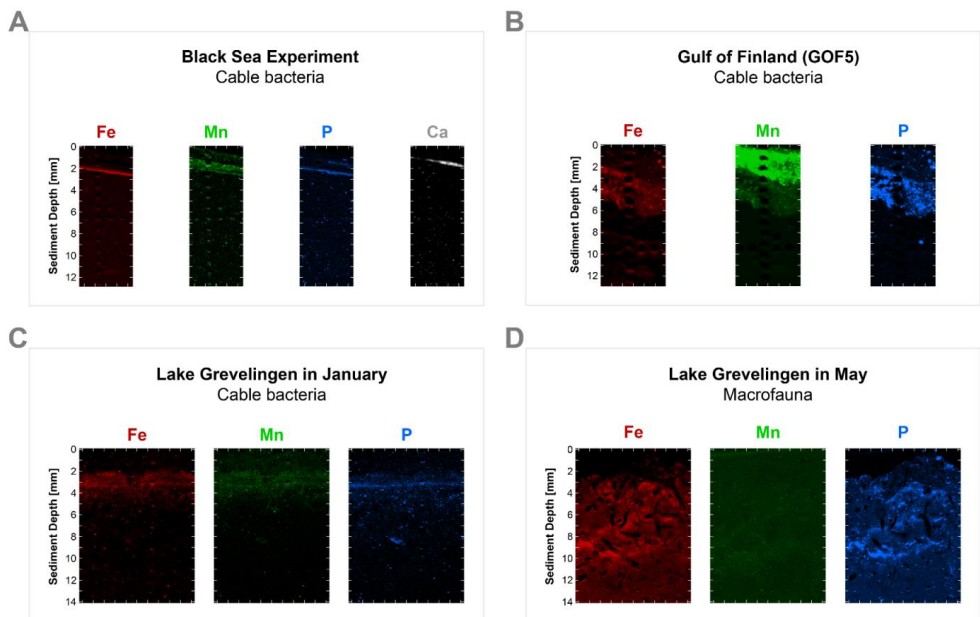

**Fig. 8.** High-resolution elemental maps of Fe (red), Mn (green), P (blue) and Ca (white) of surface sediments. These maps are shown in true vertical orientation and the colours accentuate the relative count intensities adjusted for brightness and contrast to highlight the features in the sediment. The tick marks represent 1 mm intervals. µXRF maps of the surface sediment **(A)** from the incubation experiment, **(B)** from the Gulf of Finland at site GOF5 in June (Hermans et al. Submitted), **(C)** from Lake Grevelingen in January (when cable bacteria become active) and **(D)** from Lake Grevelingen in May (showing the effects of bioturbation as described in Seitaj et al. (2015)).



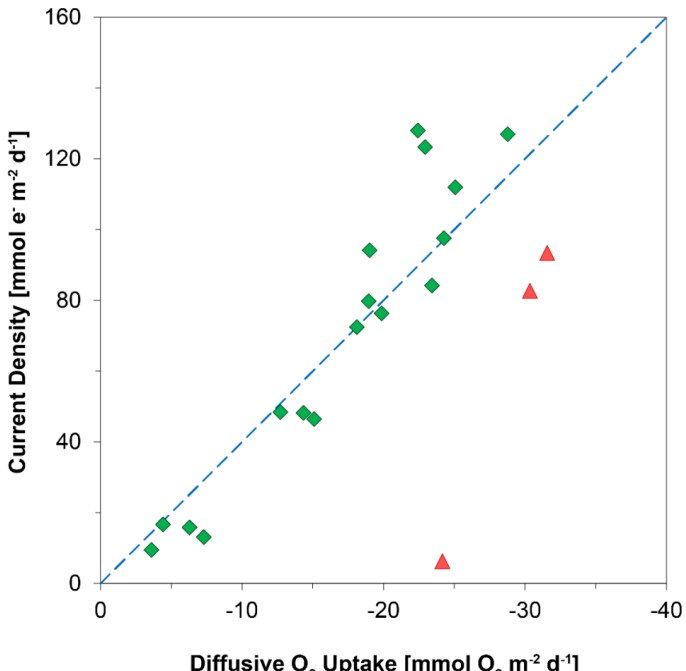

913

**Fig. 9.** The relationship between the diffusive uptake of $O_2$ (mmol $O_2$ m$^{-2}$ d$^{-1}$) and the current density of long-distance

electron transport (mmol e$^-$ m$^{-2}$ d$^{-1}$). Red triangles are data for days 0 and 5. Green diamonds are data for all other time

points. The blue line represents the expected correlation between the cathodic $O_2$ consumption rate and the current density

assuming a 1:4 ratio (Nielsen et al. 2010). Here, a positive value indicates an upward flux, whereas a negative value

represents a downward flux.






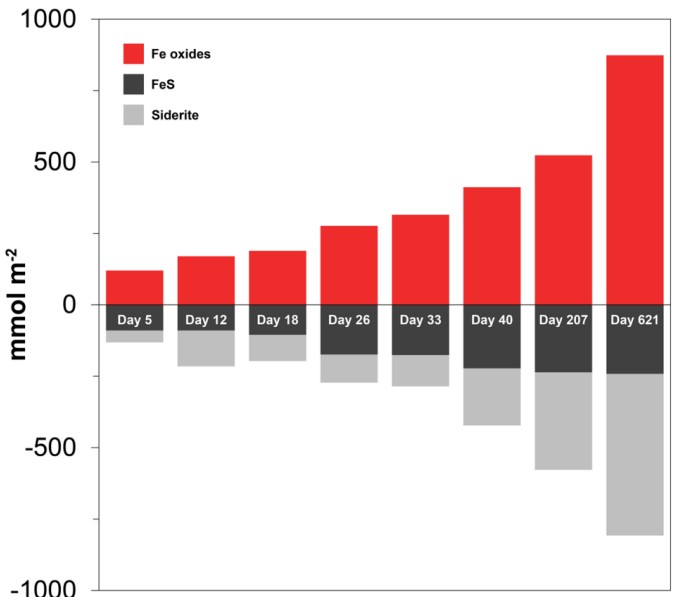


**Fig. 10.** Time-series of the depth integrated (0-5 cm) increase in Fe oxides (red) and the depletion of FeS (black) and siderite

(grey) in mmol m$^{-2}$. Negative values represent a decrease, whereas positive values indicate an increase in the mineral pools.





**Table 1.** Key site characteristics: latitude, longitude, water depth, bottom water $O_2$ concentration, *in-situ* $O_2$ uptake, *in-situ*
$O_2$ penetration depth in the sediment, porosity and salinity. These data were retrieved from Lenstra et al. (2019). Our study
site is station 9 in Lenstra et al. (2019).

| Black Sea (Station 9) | | Unit |
|---|---|---|
| Latitude | 44°34.9' | N |
| Longitude | 29°11.4' | E |
| Water depth | 27 | m |
| Bottom water $O_2$ | 92 | µM |
| $O_2$ uptake | 25.8 ± 1.77 | mmol m$^{-2}$ d$^{-1}$ |
| $O_2$ penetration depth | 2.25 | mm |
| Porosity | 0.86 | - |
| Salinity | 17.881 | - |


**Table 2.** Mass balance of $O_2$ consumption. The diffusive uptake of $O_2$ as calculated from the $O_2$ depth profiles (column 1)
was compared to the potential $O_2$ demand from the oxidation of $NH_4^+$, $Fe^{2+}$ and $Mn^{2+}$ (column 2-4). The $O_2$ consumption of
the oxidation of $NH_4^+$, $Fe^{2+}$ and $Mn^{2+}$ was determined based on the stoichiometry of $NH_4^+$, $Fe^{2+}$ and $Mn^{2+}$ oxidation with $O_2$
as described in Reed et al. (2011). The oxidation of dissolved $Fe^{2+}$ and $Mn^{2+}$ only played a minor role in the total $O_2$
consumption during the experiment, contributing only 0.9 to 3.8% and 0.1 to 0.4%, respectively.

| | $O_2$ [mmol m$^{-2}$ d$^{-1}$] | Potential $O_2$ Demand | | | e$^-$ [mmol m$^{-2}$ d$^{-1}$] |
|---|---|---|---|---|---|
| | | $NH_4^+$ [mmol m$^{-2}$ d$^{-1}$] | $Fe^{2+}$ [mmol m$^{-2}$ d$^{-1}$] | $Mn^{2+}$ [mmol m$^{-2}$ d$^{-1}$] | |
| Day 5 | -23.35 | 9.42 | 0.21 | 0.05 | 82.68 |
| Day 12 | -23.24 | 8.46 | 0.89 | 0.09 | 111.94 |
| Day 18 | -21.10 | 8.04 | 0.70 | 0.08 | 127.97 |
| Day 26 | -23.00 | 7.58 | 0.63 | 0.07 | 97.55 |
| Day 33 | -22.80 | 5.06 | 0.62 | 0.05 | 84.16 |
| Day 40 | -19.60 | 4.88 | 0.60 | 0.06 | 76.31 |
| Day 207 | -6.90 | 3.52 | 0.08 | 0.01 | 13.10 |
| Day 621 | -3.25 | N/A | 0.03 | 0.01 | 9.47 |






**Table 3.** Mass balance of Fe. Time-series of the depth integrated (0-5 cm) increase in Fe oxides and the depth integrated (0-5
cm) depletion of FeS and $FeCO_3$ (siderite) in mmol m$^{-2}$. All values are reported in mmol Fe m$^{-2}$. Negative values represent a
decrease, whereas positive values indicate an increase in the mineral pools.

| | $\Delta$Fe oxides [mmol m$^{-2}$] | $\Delta$FeS [mmol m$^{-2}$] | $\Delta FeCO_3$ [mmol m$^{-2}$] |
|---|---|---|---|
| Day 5 | 120 | -90 | -42 |
| Day 12 | 170 | -90 | -126 |
| Day 18 | 189 | -105 | -92 |
| Day 26 | 276 | -174 | -99 |
| Day 33 | 315 | -176 | -109 |
| Day 40 | 412 | -223 | -200 |
| Day 207 | 523 | -236 | -341 |
| Day 621 | 874 | -242 | -566 |


**Table 4.** Sources of $\sum H_2S$ calculated from the reduction of $SO_4^{2-}$ and the dissolution of FeS. The numbers are presented
either as mmol m$^{-2}$ d$^{-1}$ or as the relative percentage of the $\sum H_2S$ production. The amount of S from the dissolution of FeS
was estimated from the upward diffusive flux of $Fe^{2+}$ (Fig. 6C) and the relative fraction of FeS (FeS/FeS+siderite) based on
the mass balance calculations (Table 3).

| | S from $SO_4^{2-}$ reduction [mmol m$^{-2}$ d$^{-1}$] | S from FeS dissolution [mmol m$^{-2}$ d$^{-1}$] | S from $SO_4^{2-}$ reduction [%] | S from FeS dissolution [%] |
|---|---|---|---|---|
| Day 5 | 10.49 | 0.56 | 95% | 5% |
| Day 12 | 17.60 | 1.48 | 92% | 8% |
| Day 18 | 8.87 | 1.50 | 86% | 14% |
| Day 26 | 11.15 | 1.61 | 87% | 13% |
| Day 33 | 8.54 | 1.52 | 85% | 15% |
| Day 40 | 7.57 | 1.25 | 86% | 14% |
| Day 207 | 10.38 | 0.13 | 99% | 1% |
| Day 621 | 5.36 | 0.03 | 99% | 1% |
