# Peer review of "Biogeochemical Impact of Cable Bacteria on Coastal Black Sea Sediment"

_Biogeosciences, 2020_

## Referee Comment (RC1) · Anonymous Referee #1 · 13 Aug 2020

The study documents the effect of cable bacteria on sediment geochemistry in repacked sediment cores on sediment collected from the Black Sea. The findings support current paradigms in the cable bacteria literature. This is a nice case study that has been well presented and written up. The dominance of cable bacteria in the oxygen budget is an interesting finding. There is also some additional data on element mapping that sheds further light on the effect of cable bacteria on sediment geochemistry at different sites. I only have a few relatively minor comments and suggestions for improvement.

The authors used the classic sediment repacking method to start a cable growth cycle. What effect might homogenisation have had on the final findings? For example, would siderite be likely to be so close to the surface under normal circumstances.

Line 163 – no need to say 'bottom water' just water samples? Line 172 – please elaborate a little on exactly what you mean by salt corrections Line 205 – could you add a sentence on how the embedding was achieved? Line 210 onwards – consider adding this to methods also Line 261 – Only Ca and Si fluxes are presented, I couldn't see them? Line 384 not clear what you mean here, please elaborate. Line 415 – could it be that the nitrate is just denitrified?

Also on this point, it seems that not flux measurements were made for nitrate. It seems likely that some nitrate is released to the water. It might be worth a brief discussion of a few scenarios here. All the nitrate is released to the water column, all the nitrate is denitrified by sediment bacteria and all the nitrate is denitrified by cable bacteria

Line 485 – very interesting! Line 522 – not obvious to me from from fig 8A, it is interesting, can you make this clearer? Line 546 I agree this is likely driven by cables, but how is this different from a straight reaction diffusion scenario (given ubiquity of cables, such a scenario does seem unlikely though). I think this idea needs a little more development and explanation as to how it might actually be applied.

Figure 3 not clear how this was generated. Based on the pictures?

---

## Referee Comment (RC2) · Anonymous Referee #2 · 31 Aug 2020

The manuscript represents a very comprehensive study of potential processes and effects of cable bacteria in sediments. Investigations on cable bacteria and their influence on biogeochemical processes are still in the beginning, but more and more studies show their importance for the element cycling; importance of cable bacteria activity on the oxygen demand in coastal sediments. In the present study, the authors used sediment cores from the coastal area of the Black Sea, which they homogenized and freed from macrofauna. This probably increased the availability of labile organic material and its distribution in deeper sediment layers. Furthermore, the sediment was anoxically stored until the experiment, during the experiment the overlying bottom water was saturated with oxygen so that a steady state must be established at the beginning of the incubations. This fact does not reduce the results of the experiment or the quality

of the manuscript. However, the authors should consider the study presented here as potential processes and not directly related to a coastal region (in this case the Black Sea). Therefore, I would strongly suggest to rewrite the manuscript and change the focus of the manuscript by concentrating on the "potential processes and biogeochemical impacts" rather than to directly relate it to coastal sediments of the Black Sea. The difference between the natural distribution of cable bacteria and the experiment is also evident when looking at Fig. 1c. The authors can use their main results as shown here, but the focus should be on the conditions used in their experiment, which are rather artificial, but very nicely show the potential of cable bacteria in the biogeochemical cycling. In a second step the transfer to coastal sediments and their biogeochemical conditions can be done. Here the manuscript lacks the coherence (hypoxia and oxygen depletion as mentioned in the Introduction). In a final paragraph the transfer of the laboratory experiment to natural sediments and possible variations in biogeochemical processes as well as the influence of macrofauna (bioturbation and bioirrigation) can be discussed.

- Is there any information about the organic carbon content of the sediment and how this changes over the incubation period ? I would assume that this is the major driver for the development of biogeochemical zonation.

- How does the development of the oxic zone, as shown in the experiment, relate to natural variations in coastal sediments ? - How does the experiment relate to the development of hypoxia and depletion of oxygen in coastal areas ? The experiment shows the opposite reaction (from anoxic surface layer to an oxygenated layer).

- line2 121/122: ... with overlying water ... Was this bottom water taken from the site or artificial water, as used for the aquarium?

- line 153: ....... core was place outside the aquarium ..... Why was the core taken out ? was the incubation temperature maintained?

- Was the overlying water during the 24-hour incubation for the solute flux measurements stirred to avoid stratification? This could have influenced the flux across the sediment-water interface because stagnant waters lead to an increase of the Diffusive Boundary Layer, which controls the solute exchange.

- Pore water profiles (specially Fig 1a, Fig 2) are very small and it is difficult to recognize the different profiles (O2, pH, H2S) different; graphs should be enlarged.

Please also note the supplement to this comment:
https://bg.copernicus.org/preprints/bg-2020-292/bg-2020-292-RC2-supplement.pdf

---

## Author Comment (AC1) · 2 Oct 2020

The manuscript represents a very comprehensive study of potential processes and effects of cable bacteria in sediments. Investigations on cable bacteria and their influence on biogeochemical processes are still in the beginning, but more and more studies show their importance for the element cycling; importance of cable bacteria activity on the oxygen demand in coastal sediments. In the present study, the authors used sediment cores from the coastal area of the Black Sea, which they homogenized and freed from macrofauna. This probably increased the availability of labile organic material and its distribution in deeper sediment layers. Furthermore, the sediment was anoxically stored until the experiment, during the experiment the overlying bottom water was saturated with oxygen so that a steady state must be established at the beginning

of the incubations. This fact does not reduce the results of the experiment or the quality of the manuscript.

Reply: We thank Anonymous Referee #2 for reviewing our paper and the insightful and constructive feedback. Please find our replies to each comment below.

Comment #1 However, the authors should consider the study presented here as potential processes and not directly related to a coastal region (in this case the Black Sea). Therefore, I would strongly suggest to rewrite the manuscript and change the focus of the manuscript by concentrating on the "potential processes and bio geochemical impacts" rather than to directly relate it to coastal sediments of the Black Sea.

Reply: We are aware that the outcomes from our experiment are potential processes, which cannot be directly translated to the field site. This is why, in our title we specifically used the term "on coastal Black Sea sediment" instead of "in coastal Black Sea sediments". Other examples of sentences in our manuscript that indicate that we are not directly relating our results to a coastal region are:

1) Line No. 100: "In this study, we assess whether cable bacteria activity can establish in sediments that are relatively poor in FeS in an incubation experiment using siderite-bearing sediments from a coastal site in the Black Sea."

2) Line No. 552: "We can only speculate about the possible in-situ relevance of cable bacteria at the coastal site in the western Black Sea where the sediment for our incubation was collected."

3) Line No. 565-566: "Further field studies are required to assess the role of cable bacteria at our field site, preferably including an assessment of the burrow structures."

We will add the following additional text in the abstract, introduction and conclusion sections to further emphasise that we are referring to potential processes.

Abstract: "to determine the potential impact of their activity on the cycling of Fe, phosphorus (P) and sulphur (S)."

Introduction: "In this study, we assess whether cable bacteria activity can establish and thrive in sediments that are relatively poor in FeS. Although, this will be done in a controlled incubation experiment with siderite-bearing sediments from a coastal site in the Black Sea, our findings are relevant for natural environments populated by cable bacteria."

Conclusion: "The results of our laboratory incubation (with a total duration of 621 days) show that cable bacteria can potentially strongly impact the Fe, Mn, P and S dynamics in coastal sediments. The strong acidity of the pore water associated with the activity of cable bacteria, which was monitored using microsensor profiling of the EP during the experiment, led to dissolution of FeS and siderite and formation of Fe and Mn oxides and Ca-P in mineral form near the sediment surface. Our experimental results provide conclusive evidence for siderite dissolution driven by cable bacteria activity as a source of Fe that can form an Fe oxide-enriched surface layer."

Comment #2 The difference between the natural distribution of cable bacteria and the experiment is also evident when looking at Fig. 1c. The authors can use their main results as shown here, but the focus should be on the conditions used in their experiment, which are rather artificial, but very nicely show the potential of cable bacteria in the biogeochemical cycling.

Reply: See our reply to comment #1 and the associated changes in the text. We are aware that the outcomes of our experiment regarding the impact of cable bacteria are amplified when compared to field conditions and that we cannot directly link this to the field site. This is because we optimized conditions to sustain metabolic activity of cable bacteria, and their growth:

1) The sediment was homogenised, which is known to induce growth of cable bacteria.

2) There was no bioturbation by meiofauna and macrofauna.

3)The bottom water was continuously oxygenated.

This optimisation was deliberately done to study the effects of cable bacteria on sediment biogeochemistry. We note that the approach used is common in studies of the biogeochemical impact of cable bacteria (e.g. Risgaard-Petersen et al. 2012; Rao et al. 2016).

Comment #3 In a second step the transfer to coastal sediments and their biogeochemical conditions can be done. Here the manuscript lacks the coherence (hypoxia and oxy-gen depletion as mentioned in the Introduction). In a final paragraph the transfer of the laboratory experiment to natural sediments and possible variations in biogeochemical processes as well as the influence of macrofauna (bioturbation and bioirrigation) can be discussed.

Reply: Sections 4.1 to 4.4 focus only on the experiment. In sections 4.5 and 4.6 we discuss the implications for the field. We will add text to section 4.5 and 4.6 to clarify when we are referring to other laboratory experiments and results of field studies and the link with hypoxia.

1) Line 507-509: "This colour zonation is typical for sediments that have been geochemically affected by cable bacteria activity, as seen both in laboratory experiments (Nielsen and Risgaard-Petersen 2015) and at coastal field sites (Sulu-Gambari et al. 2016)"

2) Line 546-548: "may act as an additional sediment marker for present or recent cable bacteria activity, both in laboratory experiments and at field sites, also in cases where visual observations are not conclusive."

3) Line 548: "Macrofaunal activity within natural environments likely counteracts or prevents strong focusing of Fe oxides and associated P within such a thin subsurface layer."

4) Line 553-554: "At this site, which is in a region that is subject to seasonal hypoxia (Capet et al. 2013), both bivalves (up to ∼7200 ind. m-2) and polychaetes (up to

∼1700 ind. m-2). . ."

Comment #4 Is there any information about the organic carbon content of the sediment and how this changes over the incubation period? I would assume that this is the major driver for the development of biogeochemical zonation.

Reply: We now include the organic carbon content of the upper 0-10 cm of the sediment at the field site, as determined by Lenstra et al. (2019), in Table 1. We did not determine the change in organic carbon contents during the experiment because, at the typical rates of organic matter degradation expected here, only a small change in organic carbon content would be observed. Hence, we chose to focus on pore water NH4+ profiles to obtain insight in rates of (anaerobic) organic matter degradation.

Comment #5 How does the development of the oxic zone, as shown in the experiment, relate to natural variations in coastal sediments ?

Reply: The range in O2 penetration observed in the experiment is comparable to that observed in coastal systems with seasonal hypoxia (e.g. Seitaj et al. 2015). We will include this in the manuscript: Line No. 514-516: "During the experiment, O2 penetration varied within a narrow range and was initially fixed between 1 and 2 mm depth (Fig. 3A), with the layer highly enriched in Fe forming mostly at a depth of 2 mm (Fig. 8A). Such a range in O2 penetration is in accordance with observations in coastal sediments (e.g. Seitaj et al. 2015). The formation of the Fe-enriched layer can be explained by. . ."

Comment #6 How does the experiment relate to the development of hypoxia and depletion of oxygen in coastal areas ? The experiment shows the opposite reaction (from anoxic surface layer to an oxygenated layer).

Reply: As shown in previous work that we refer to in our manuscript (Seitaj et al. 2015; Sulu-Gambari et al. 2016), cable bacteria can induce formation of Fe and Mn oxides in seasonally hypoxic coastal systems during periods of oxygenated bottom waters in

spring. As explained in line No. 41-44, the presence of these Fe and Mn oxides can delay the transition towards euxinia by removing hydrogen sulfide. Our experimental setting can be compared to the onset of bottom water re-oxygenation in spring in such seasonally hypoxic environments (i.e. the sediment was stored anoxically, and then exposed to oxygenated water). This allows study of the mineral dissolution and formation reactions in sediments populated by cable bacteria under such conditions, which is the goal of this work.

Comment #7 line2 121/122: .... with overlying water ....Was this bottom water taken from the site or artificial water, as used for the aquarium?

Reply: The 20 cm long cores (filled with 15 cm sediment) were gently submerged in the two aquaria. Hence, the overlying water in the cores at the start of each incubation was the same as the artificial water used in the aquaria.

Comment #8 line 153: ....... core was place outside the aquarium .....Why was the core taken out ? was the incubation temperature maintained?

Reply: The entire experiment was carried out at in a temperature controlled laboratory at 20 °C. We will make explicit that the flux incubations also took place at 20 °C: "At each time point, one core was placed outside the aquarium at 20 °C..."

Comment #9 Was the overlying water during the 24-hour incubation for the solute flux measurements stirred to avoid stratification? This could have influenced the flux across the sediment-water interface because stagnant waters lead to an increase of the Diffusive Boundary Layer, which controls the solute exchange.

Reply: The overlying water was mixed continuously by bubbling it continuously with air. The airflow was set in such a way that the water was effectively mixed, while the surface layer of the sediment was left undisturbed. We will make this more explicit in the methods: "At each time point, one core was placed outside the aquarium at 20 °C, and the isolated volume of overlying water in the core was continuously aerated. Potential

stratification of the overlying water was prevented by actively bubbling it. Parafilm was wrapped on top of the cores to prevent evaporation."

Comment #10 Pore water profiles (especially Fig 1a, Fig 2) are very small and it is difficult to recognize the different profiles (O2, pH, H2S) different; graphs should be enlarged.

Reply: We enlarged these graphs and we increased the font sizes to improve the readability of the figures.

References

Bockris, J. O. M., and Reddy, A. K. (1998). Ion-ion interactions. Modern Electrochemistry 1: Ionics: 225-359.

Capet, A., Beckers, J.-M., and Grégoire, M. (2013). Drivers, mechanisms and long-term variability of seasonal hypoxia on the Black Sea northwestern shelf–is there any recovery after eutrophication. Biogeosciences 10: 3943-3962.

Hermans, M., Astudillo Pascual, M., Behrends, T., Lenstra, W. K., Conley, D. J., and Slomp, C. P. (Submitted). Coupled dynamics of iron, manganese and phosphorus in brackish coastal sediments populated by cable bacteria.

Jilbert, T., de Lange, G., and Reichart, G. J. (2008). Fluid displacive resin embedding of laminated sediments: preserving trace metals for high‐resolution paleoclimate investigations. Limnology and Oceanography: Methods 6: 16-22.

Jilbert, T., and Slomp, C. P. (2013). Iron and manganese shuttles control the formation of authigenic phosphorus minerals in the euxinic basins of the Baltic Sea. Geochimica et Cosmochimica Acta 107: 155-169.

Lenstra, W. and others (2019). The shelf-to-basin iron shuttle in the Black Sea revisited. Chemical Geology 511: 314-341.

Marzocchi, U. and others (2014). Electric coupling between distant nitrate reduction

and sulfide oxidation in marine sediment. The ISME journal 8: 1682-1690.

Nielsen, L. P., and Risgaard-Petersen, N. (2015). Rethinking sediment biogeochemistry after the discovery of electric currents. Annual review of marine science 7: 425-442.

Rao, A. M., Malkin, S. Y., Hidalgo-Martinez, S., and Meysman, F. J. (2016). The impact of electrogenic sulfide oxidation on elemental cycling and solute fluxes in coastal sediment. Geochimica et Cosmochimica Acta 172: 265-286.

Risgaard-Petersen, N., Revil, A., Meister, P., and Nielsen, L. P. (2012). Sulfur, iron-, and calcium cycling associated with natural electric currents running through marine sediment. Geochimica et Cosmochimica Acta 92: 1-13.

Seitaj, D. and others (2015). Cable bacteria generate a firewall against euxinia in seasonally hypoxic basins. Proceedings of the National Academy of Sciences 112: 13278-13283.

Sulu-Gambari, F. and others (2018). Phosphorus cycling and burial in sediments of a seasonally hypoxic Marine Basin. Estuaries and Coasts 41: 921-939.

Sulu-Gambari, F., Seitaj, D., Behrends, T., Banerjee, D., Meysman, F. J., and Slomp, C. P. (2016). Impact of cable bacteria on sedimentary iron and manganese dynamics in a seasonally-hypoxic marine basin. Geochimica et Cosmochimica Acta 192: 49-69.

**A**

[Figure]

**B**

**C**

Volumetric Density [m cm⁻³]

[Figure]

**D**

**Fig. 1.** Figure 1

[Figure]

**Fig. 2.** Figure 2

**Day 5** | **Day 12** | **Day 18** | **Day 26** | **Day 33** | **Day 40** | **Day 207** | **Day 621** | **Control**

$NH_4^+$ [mM]

$SO_4^{2-}$ [mM]

$Fe^{2+}$ [mM]

$Mn^{2+}$ [mM]

$Ca^{2+}$ [mM]

$HPO_4^{2-}$ [mM]

$H_4SiO_4$ [mM]

**Fig. 3.** Figure 5

---

## Author Comment (AC2) · 2 Oct 2020

The study documents the effect of cable bacteria on sediment geochemistry in repacked sediment cores on sediment collected from the Black Sea. The findings support current paradigms in the cable bacteria literature. This is a nice case study that has been well presented and written up. The dominance of cable bacteria in the oxygen budget is an interesting finding. There is also some additional data on element mapping that sheds further light on the effect of cable bacteria on sediment geochemistry at different sites. I only have a few relatively minor comments and suggestions for improvement.

Reply: We thank Anonymous Referee #1 for reviewing our paper and the insightful and

constructive feedback. Please find our replies to each comment below.

Comment #1 The authors used the classic sediment repacking method to start a cable growth cycle. What effect might homogenisation have had on the final findings? For example, would siderite be likely to be so close to the surface under normal circumstances.

Reply: The homogenisation likely did not have a significant effect on our final findings. Homogenisation of the sediment is essential to obtain proper replicate cores, and is known to induce growth of cable bacteria. While some of the FeS and siderite might have oxidised during sieving, homogenising and repacking, this does not affect the conclusions of our experiment. There is still ample FeS and siderite present in our sediment cores at the start of the experiment, and our time-series indicate that both FeS and siderite were dissolved by cable bacteria activity over time. Both FeS and siderite are frequently observed in marine surface sediments (e.g. Sulu-Gambari et al. 2016).

Comment #2 Line 163 – no need to say 'bottom water' just water samples?

Reply: Here, we specifically use the term bottom water samples, since it refers to the overlying water in the cores. Water samples could also refer to other things, such as water column or pore water samples. Therefore, we think it is best to use the term bottom water samples, to avoid potential confusion.

Comment #3 Line 172 – please elaborate a little on exactly what you mean by salt corrections.

Reply: Freeze-drying removes the water from samples. However, the salt stays behind in the dried sediment. Hence, the weight of the freeze-dried sediment used for the sequential extractions needs to be corrected for this salt content if we wish to calculate elemental/mineral concentrations, such as Fe oxide and metal bound P contents per gram of sediment. Without the salt correction the absolute elemental/mineral concentrations would be underestimated. Hence, we subtract the weight of the salt from the freeze-dried sediment to calculate the 'real' weight of the dry sediment. We will explain this in a bit more detail in our methods: "After freeze-drying, the salt from sea-water stays behind in the solid-phase fraction. To determine the actual weight of the dry sediment, it is necessary to subtract the weight of the salt from the total weight of freeze-dried sediment."

Comment #4 Line 205 – could you add a sentence on how the embedding was achieved?

Reply: The resin embedding process will be described in more detail: "On day 47, an undisturbed core (first 5 cm of surface sediment) was sampled for epoxy resin embedding for high-resolution elemental mapping (Jilbert et al. 2008; Jilbert and Slomp 2013). Sediment was carefully pushed upwards from the experimental core into a shorter (7 cm length; 1 cm diameter) mini core. This mini sub-core was then transferred to an acetone bath in a argon-filled glovebox and subsequently embedded with Spurr's epoxy resin as described in Jilbert et al. (2008). After curing, the epoxy-embedded core was split vertically using a rock saw."

Comment #5 Line 210 onwards – consider adding this to methods.

Reply: The methods used to obtain these elemental maps from the epoxy resin embedded surface sediments from the Gulf of Finland and Lake Grevelingen are described in detail in the other studies cited (Sulu-Gambari et al. 2016; Sulu-Gambari et al. 2018; Hermans et al. Submitted). Therefore, we prefer not to describe the sampling process of those resin embedded cores in section 2.4 of our methodology.

Comment #6 also Line 261 – Only Ca and Si fluxes are presented, I couldn't see them?

Reply: These Ca and Si fluxes are presented in Fig. S10 and Table S4 in the Supplementary Information, see line No. 480 in the manuscript.

Comment #7 Line 384 not clear what you mean here, please elaborate.

[Figure]

Reply: We have used Fick's law for the calculation of the diffusive fluxes, which does not take the effect of the electric field generated by cable bacteria on the diffusion potential into account. The Nernst-Planck equation, however, extends Fick's law, because solutes can also be moved with respect to the fluid by electrostatic forces. In the revised version, we will explain this in greater detail including the effect on the $SO_4^{2-}$-flux.

Comment #8 Line 415 – could it be that the nitrate is just denitrified? Also on this point, it seems that not flux measurements were made for nitrate. It seems likely that some nitrate is released to the water. It might be worth a brief discussion of a few scenarios here. All the nitrate is released to the water column, all the nitrate is denitrified by sediment bacteria and all the nitrate is denitrified by cable bacteria.

Reply: Unfortunately, we do not have data for $NO_3^-$. The problem of the abovementioned scenarios is that, if we would include the role of other groups of bacteria, and the potential release of $NO_3^-$ to the water column, the mass balance for $O_2$ would have an even greater mismatch. When looking at the stoichiometry of $NO_3^-$ by cable bacteria and the conversion of N to $N_2$, we cannot close the budget fully that way. We will modify the text to explain this in more detail: "These findings can be explained, however, if we assume that at least part of the $NO_3^-$ that is being formed near the sediment-water interface is also used for the metabolic activity of cable bacteria. It has been shown that cable bacteria can couple the oxidation of 2S to $NO_3^-$ in the absence of $O_2$ (Marzocchi et al. 2014). Our data suggest that this process may also occur in sediments where $O_2$ is present in concert with $NO_3^-$ near the sediment-water interface. However, we cannot exclude release of $NO_3^-$ to the water column or denitrification by other bacteria in the sediment."

Comment #9 Line 485 – very interesting!

Reply: Thank you, this potential niche for vivianite formation is indeed an interesting finding.

Comment #10 Line 522 – not obvious to me from Fig 8A, it is interesting, can you make this clearer?

Reply: We will make this more explicit in the text: "While the Fe oxide layer is clearly enriched in P, we also observed a second layer enriched in P very close to the sediment-water interface (Fig. 8A). This layer is located above the Fe oxide layer, and in this layer P is strongly correlated with Ca."

Comment #11 Line 546 I agree this is likely driven by cables, but how is this different from a straight reaction diffusion scenario (given ubiquity of cables, such a scenario does seem unlikely though). I think this idea needs a little more development and explanation as to how it might actually be applied.

Reply: Focusing of Fe and Mn oxides and associated P can indeed also occur in sediments overlain by oxic waters, where no cable bacteria are active. However, as demonstrated by our experiment (and various other studies (e.g. Risgaard-Petersen et al. 2012; Rao et al. 2016), in sediment populated by active cable bacteria, the upward fluxes of Fe2+ and Mn2+ are higher due to the dissolution of FeS, Fe- and Mn carbonates. This allows strong focusing of Fe and Mn oxides in a thin layer within a relatively short time frame. We will add the following section in our manuscript: "Focussing of Fe and Mn oxides in the surface sediment is not exclusively tied to the activity of cable bacteria, and can also occur in the absence of cable bacteria. However, the upward fluxes of Fe2+ and Mn2+ in sediments populated by cable bacteria are higher due to active dissolution of Fe and Mn minerals at depth (e.g. Risgaard-Petersen et al. 2012; Rao et al. 2016). Hence, within the same time period following an environmental perturbation (such as a transition to oxic bottom waters after a period of anoxia or mixing of the sediment), more Fe2+ and Mn2+ can oxidise upon contact with O2 near the sediment-water interface and thus stronger enrichments of Fe and Mn minerals will be observed. Therefore, focussing of Fe and Mn oxides in subsurface sediments is likely more prominent and stronger in sediments populated by active cable bacteria compared to sediments where no cable bacteria are active under such conditions."

Comment #12 Figure 3 not clear how this was generated. Based on the pictures?

Reply: The depth intervals of the oxic, suboxic and anoxic zone are based on the micro-electrode data. We will make it more explicit in the caption of Fig. 3.: "Time-series of the development of the oxic zone (orange), suboxic zone (light grey) and the anoxic/sulphidic zone (dark grey) in the sediment. These zones were calculated from 3 replicate microelectrode depth profiles retrieved from two different cores."

References

Hermans, M., Astudillo Pascual, M., Behrends, T., Lenstra, W. K., Conley, D. J., and Slomp, C. P. (Submitted). Coupled dynamics of iron, manganese and phosphorus in brackish coastal sediments populated by cable bacteria.

Jilbert, T., de Lange, G., and Reichart, G. J. (2008). Fluid displacive resin embedding of laminated sediments: preserving trace metals for high‐resolution paleoclimate investigations. Limnology and Oceanography: Methods 6: 16-22.

Jilbert, T., and Slomp, C. P. (2013). Iron and manganese shuttles control the formation of authigenic phosphorus minerals in the euxinic basins of the Baltic Sea. Geochimica et Cosmochimica Acta 107: 155-169.

Marzocchi, U. and others (2014). Electric coupling between distant nitrate reduction and sulfide oxidation in marine sediment. The ISME journal 8: 1682-1690.

Rao, A. M., Malkin, S. Y., Hidalgo-Martinez, S., and Meysman, F. J. (2016). The impact of electrogenic sulfide oxidation on elemental cycling and solute fluxes in coastal sediment. Geochimica et Cosmochimica Acta 172: 265-286.

Risgaard-Petersen, N., Revil, A., Meister, P., and Nielsen, L. P. (2012). Sulfur, iron-, and calcium cycling associated with natural electric currents running through marine sediment. Geochimica et Cosmochimica Acta 92: 1-13.

Sulu-Gambari, F. and others (2018). Phosphorus cycling and burial in sediments of a

seasonally hypoxic Marine Basin. Estuaries and Coasts 41: 921-939.

Sulu-Gambari, F., Seitaj, D., Behrends, T., Banerjee, D., Meysman, F. J., and Slomp, C. P. (2016). Impact of cable bacteria on sedimentary iron and manganese dynamics in a seasonally-hypoxic marine basin. Geochimica et Cosmochimica Acta 192: 49-69.
* * *

---

## Author Response (AR2)

**21 October 2020**

Dear Prof. Treude,

We now included the University of Helsinki in the affiliations, since I currently work here as a postdoctoral researcher and I forgot to include this in the version I submitted on October 19th. We also added more details to the other affiliations. Because of this, the line numbers that we refer to in our replies have been updated again in this version.

Kind regards,

Martijn Hermans

**19 October 2020**

Dear Prof. Treude,

We would like to thank you and the two anonymous referees for reviewing our paper and the insightful and constructive feedback. Please find our replies below where we explain how we addressed each comment and the changes that were made in our manuscript. Unless otherwise indicated, the line numbers in our replies refer to the revised manuscript. We include a marked-up version of our manuscript that highlights all the relevant changes that were made, and we now uploaded the high-resolution version of the figures at the end of our manuscript.

Kind regards,

Martijn Hermans

Prof. Treude's comments:

Instead of 'bottom water' you could also use the term "supernatant" or 'bottom-near" water. I was in the past criticized by physical oceanographers for using the term 'bottom water' in my sediment studies, because it is a fixed term in oceanography and covers a large volume (an entire water mass).

**Reply:** We prefer to use "bottom water" since for chemical oceanographers this is a common term. We now made more explicit what bottom water refers to see line **No. 35**: "Depletion of oxygen ($O_2$) in bottom waters (i.e. water directly above the seafloor)"

- The term "suboxic" opposite to oxic and anoxic is not ideal, because suboxic is also anoxic. But I am aware that the term is frequently used together with cable bacteria biogeochemistry. I am OK with keeping the term, but please double check that your final manuscript has a proper definition of the terms at the beginning (oxic = with $O_2$, suboxic = anoxic but not sulfidic, anoxic = anoxic and sulfidic).

**Reply:** We explain this in more detail in the revised version of our manuscript, see line **No. 54-55**: "This spatial coupling of surficial $O_2$ reduction with $H_2S$ oxidation at several centimetres depth creates a suboxic zone that is devoid of any $O_2$ and $H_2S$"

- In Fig. 2B a red and a green profile are shown in the same graphs. Please use a different color to allow green-red handicapped people to enjoy it.

**Reply:** We updated the figure and changed the colours to blue and orange, so it can be read by colour blind people as well.

**Anonymous Referee #1**

The study documents the effect of cable bacteria on sediment geochemistry in repacked sediment cores on sediment collected from the Black Sea. The findings support current paradigms in the cable bacteria literature. This is a nice case study that has been well presented and written up. The dominance of cable bacteria in the oxygen budget is an interesting finding. There is also some additional data on element mapping that sheds further light on the effect of cable bacteria on sediment geochemistry at different sites. I only have a few relatively minor comments and suggestions for improvement.

**Reply:** We thank Anonymous Referee #1 for reviewing our paper and the insightful and constructive feedback. Please find our replies to each comment below.

**Comment #1** The authors used the classic sediment repacking method to start a cable growth cycle. What effect might homogenisation have had on the final findings? For example, would siderite be likely to be so close to the surface under normal circumstances.

**Reply:** The homogenisation likely did not have a significant effect on our final findings. Homogenisation of the sediment is essential to obtain proper replicate cores and is known to induce growth of cable bacteria. While some of the FeS and siderite might have oxidised during sieving, homogenising and repacking, this does not affect the conclusions of our experiment. There is still ample FeS and siderite present in our sediment cores at the start of the experiment, and our time-series indicate that both FeS and siderite were dissolved by cable bacteria activity over time. Both FeS and siderite are frequently observed in marine surface sediments (e.g. Sulu-Gambari et al. 2016).

**Comment #2** Line 163 – no need to say 'bottom water' just water samples?

**Reply:** Here, we specifically use the term bottom water samples, since it refers to the overlying water in the cores. Water samples could also refer to water column or pore water samples. Therefore, we think it is best to use the term bottom water samples, to avoid potential confusion.

**Comment #3** Line 172 – please elaborate a little on exactly what you mean by salt corrections.

**Reply:** Freeze-drying removes the water from samples. However, the salt stays behind in the dried sediment. Hence, the weight of the freeze-dried sediment used for the sequential extractions needs to be corrected for this salt content if we wish to calculate elemental/mineral concentrations, such as Fe oxide and metal bound P contents per gram of sediment. Without the salt correction the absolute elemental/mineral concentrations would be underestimated. Hence, we subtract the weight of the salt from the freeze-dried sediment to calculate the 'real' weight of the dry sediment. This is now explained in more detail in our methods, see line **No. 178-181**: "After freeze-drying, the salt from seawater stays behind in the solid-phase fraction. To determine the actual weight of the dry sediment, it is necessary to subtract the weight of the salt from the total weight of freeze-dried sediment."

**Comment #4** Line 205 – could you add a sentence on how the embedding was achieved?

**Reply:** The resin embedding process is described in more detail, see line **No. 212-217**: "On day 47, an undisturbed core (first 5 cm of surface sediment) was sampled for epoxy resin embedding for high-resolution elemental mapping (Jilbert et al. 2008; Jilbert and Slomp 2013). Sediment was carefully pushed upwards from the experimental core into a shorter (7 cm length; 1 cm diameter) mini core. This mini sub-core was then transferred to an acetone bath in a argon-filled glovebox and subsequently embedded with Spurr's epoxy resin as described in Jilbert et al. (2008). After curing, the epoxy-embedded core was split vertically using a rock saw."

**Comment #5** Line 210 onwards – consider adding this to methods.

**Reply:** The methods used to obtain these elemental maps from the epoxy resin embedded surface sediments from the Gulf of Finland and Lake Grevelingen are described in detail in the other studies cited (Sulu-Gambari et al. 2016; Sulu-Gambari et al. 2018; Hermans et al. Submitted). Therefore, we prefer not to describe the sampling process of those resin embedded cores in section 2.4 of our methodology.

**Comment #6** also Line 261 – Only Ca and Si fluxes are presented, I couldn't see them?

**Reply:** These Ca and Si fluxes are presented in Fig. S10 and Table S4 in the Supplementary Information, see line **No. 488** in the manuscript.

**Comment #7** Line 384 not clear what you mean here, please elaborate.

**Reply:** We have used Fick's law for the calculation of the diffusive fluxes, which not does take the effect of the electric field generated by cable bacteria on the diffusion potential into account. The Nernst-Planck equation, however, extends Fick's law, because solutes can also be moved with respect to the fluid by electrostatic forces. By using Fick's law, we underestimate the $SO_4^{2-}$ reduction rate by a most ~10-20%. We will describe this in more detail, see line **No. 396-400**: "Solutes can also move with respect to the fluid by electrostatic forces (Bockris and Reddy 1998). Given the relatively low strength of the electric field in the cores ($<0.073V$ m$^{-1}$ at day 18; as estimated from Fig 2B), including the contribution of ionic drift to the sulphate flux would lead to $SO_4^{2-}$ reduction rates that are at most 10-20% higher."

**Comment #8** Line 415 – could it be that the nitrate is just denitrified? Also on this point, it seems that not flux measurements were made for nitrate. It seems likely that some nitrate is released to the water. It might be worth a brief discussion of a few scenarios here. All the nitrate is released to the water column, all the nitrate is denitrified by sediment bacteria and all the nitrate is denitrified by cable bacteria.

**Reply:** Unfortunately, we do not have data for $NO_3^-$. The problem of the abovementioned scenarios is that, if we would include the role of other groups of bacteria, and the potential release of $NO_3^-$ to the water column, the mass balance for $O_2$ would have an even greater mismatch. When looking at the stoichiometry of $NO_3^-$ by cable bacteria and the conversion of N to $N_2$, we cannot close the budget fully that way. The text is now modified and explains this in more detail, see line **No. 428-433**: "These findings can be explained, however, if we assume that at least part of the $NO_3^-$ that is being formed near the sediment-water interface is also used for the metabolic activity of cable bacteria. It has been shown that cable bacteria can couple the oxidation of $\sum H_2S$ to $NO_3^-$ in the absence of $O_2$ (Marzocchi et al. 2014). Our data suggest that this process may also occur in sediments where $O_2$ is present in concert with $NO_3^-$ near the sediment-water interface. However, we cannot exclude release of $NO_3^-$ to the water column or denitrification by other bacteria in the sediment."

**Comment #9** Line 485 – very interesting!

**Reply:** Thank you, this potential niche for vivianite formation is indeed an interesting finding.

**Comment #10** Line 522 – not obvious to me from Fig 8A, it is interesting, can you make this clearer?

**Reply:** This is now more explicit in the text, see line **No. 528-531**: "While the Fe oxide layer is clearly enriched in P, we also observed a second layer enriched in P very close to the sediment-water interface (Fig. 8A). This layer is located above the Fe oxide layer, and in this layer P is strongly correlated with Ca."

**Comment #11** Line 546 I agree this is likely driven by cables, but how is this different from a straight reaction diffusion scenario (given ubiquity of cables, such a scenario does seem unlikely though). I think this idea needs a little more development and explanation as to how it might actually be applied.

**Reply:** Focusing of Fe and Mn oxides and associated P can indeed also occur in sediments overlain by oxic waters, where no cable bacteria are active. However, as demonstrated by our experiment (and various other studies (e.g. Risgaard-Petersen et al. 2012; Rao et al. 2016), in sediment populated by active cable bacteria, the upward fluxes of $Fe^{2+}$ and $Mn^{2+}$ are higher due to the dissolution of FeS, Fe- and Mn carbonates. This allows strong focusing of Fe and Mn oxides in a thin layer within a relatively short time frame. We added the following section in our manuscript, see line **No. 571-581**: "Focussing of Fe and Mn oxides in the surface sediment is not exclusively tied to the activity of cable bacteria, and can also occur in the absence of cable bacteria. However, the upward fluxes of $Fe^{2+}$ and $Mn^{2+}$ in sediments populated by cable bacteria are higher due to active dissolution of Fe and Mn minerals at depth (e.g. Risgaard-Petersen et al. 2012; Rao et al. 2016). Hence, within the same time frame following an environmental perturbation (such as a transition to oxic bottom waters after a period of anoxia or mixing of the sediment), more $Fe^{2+}$ and $Mn^{2+}$ can oxidise upon contact with $O_2$ near the sediment-water interface and thus stronger enrichments of Fe and Mn minerals will be observed. Hence, focussing of Fe and Mn oxides in subsurface sediments is likely more prominent and stronger in sediments populated by active cable bacteria compared to sediments where no cable bacteria are active under such conditions."

**Comment #12** Figure 3 not clear how this was generated. Based on the pictures?

**Reply:** The depth intervals of the oxic, suboxic and anoxic zone are based on the micro-electrode data. We made this more explicit in the caption of Fig. 3., see line **No. 924-926**: "Time-series of the development of the oxic zone (orange), suboxic zone (light grey) and the anoxic/sulphidic zone (dark grey) in the sediment. These zones were calculated from 3 replicate microelectrode depth profiles retrieved from two different cores."

**Anonymous Referee #2**

The manuscript represents a very comprehensive study of potential processes and effects of cable bacteria in sediments. Investigations on cable bacteria and their influence on biogeochemical processes are still in the beginning, but more and more studies show their importance for the element cycling; importance of cable bacteria activity on the oxygen demand in coastal sediments. In the present study, the authors used sediment cores from the coastal area of the Black Sea, which they homogenized and freed from macrofauna. This probably increased the availability of labile organic material and its distribution in deeper sediment layers. Furthermore, the sediment was anoxically stored until the experiment, during the experiment the overlying bottom water was saturated with oxygen so that a steady state must be established at the beginning of the incubations. This fact does not reduce the results of the experiment or the quality of the manuscript.

**Reply:** We thank Anonymous Referee #2 for reviewing our paper and the insightful and constructive feedback. Please find our replies to each comment below.

**Comment #1** However, the authors should consider the study presented here as potential processes and not directly related to a coastal region (in this case the Black Sea). Therefore, I would strongly suggest to rewrite the manuscript and change the focus of the manuscript by concentrating on the "potential processes and bio geochemical impacts" rather than to directly relate it to coastal sediments of the Black Sea.

**Reply:** We are aware that the outcomes from our experiment are potential processes, which cannot be directly translated to the field site. This is why, in our title we specifically used the term "on coastal Black Sea sediment" instead of "in coastal Black Sea sediments". Other examples of sentences in our manuscript (here these line numbers refer to the old version of our manuscript) that indicate that we are not directly relating our results to a coastal region are:

- Line **No. 100**: "In this study, we assess whether cable bacteria activity can establish in sediments that are relatively poor in FeS in an incubation experiment using siderite-bearing sediments from a coastal site in the Black Sea."
- Line **No. 552**: "We can only speculate about the possible *in-situ* relevance of cable bacteria at the coastal site in the western Black Sea where the sediment for our incubation was collected."
- Line **No. 565-566**: "Further field studies are required to assess the role of cable bacteria at our field site, preferably including an assessment of the burrow structures."

We added the following additional text in the abstract, introduction and conclusion sections to further emphasise that we are referring to potential processes.

Abstract; line **No. 17-18**: "to determine the potential impact of their activity on the cycling of Fe, phosphorus (P) and sulphur (S)."

Introduction; line **No. 103-106**: "In this study, we assess whether cable bacteria activity can establish and thrive in sediments that are relatively poor in FeS. Although, this will be done in a controlled incubation experiment with siderite-bearing sediments from a coastal site in the Black Sea, our findings are relevant for natural environments populated by cable bacteria."

Conclusion; line **No. 603-609**: "The results of our laboratory incubation (with a total duration of 621 days) show that cable bacteria can potentially strongly impact the Fe, Mn, P and S dynamics in coastal sediments. The strong acidity of the pore water associated with the activity of cable bacteria, which was monitored using microsensor profiling of the EP during the experiment, led to dissolution of FeS and siderite and formation of Fe and Mn oxides and Ca-P in mineral form near the sediment surface. Our experimental results provide conclusive evidence for siderite dissolution driven by cable bacteria activity as a source of Fe that can form an Fe oxide-enriched surface layer."

**Comment #2** The difference between the natural distribution of cable bacteria and the experiment is also evident when looking at Fig. 1c. The authors can use their main results as shown here, but the focus should be on the conditions used in their experiment, which are rather artificial, but very nicely show the potential of cable bacteria in the biogeochemical cycling.

**Reply:** See our reply to comment #1 and the associated changes in the text. We are aware that the outcomes of our experiment regarding the impact of cable bacteria are amplified when compared to field conditions and that we cannot directly link this to the field site. This is because we optimized conditions to sustain metabolic activity of cable bacteria, and their growth:

1) The sediment was homogenised, which is known to induce growth of cable bacteria.
2) There was no bioturbation by meiofauna and macrofauna.
3) The bottom water was continuously oxygenated.

This optimisation was deliberately done to study the effects of cable bacteria on sediment biogeochemistry. We note that the approach used is common in studies of the biogeochemical impact of cable bacteria (e.g. Risgaard-Petersen et al. 2012; Rao et al. 2016).

**Comment #3** In a second step the transfer to coastal sediments and their biogeochemical conditions can be done. Here the manuscript lacks the coherence (hypoxia and oxy-gen depletion as mentioned in the Introduction). In a final paragraph the transfer of the laboratory experiment to natural sediments and possible variations in biogeochemical processes as well as the influence of macrofauna (bioturbation and bioirrigation) can be discussed.

**Reply:** Sections 4.1 to 4.4 focus only on the experiment. In sections 4.5 and 4.6 we discuss the implications for the field. We added text to section 4.5 and 4.6 to clarify when we are referring to other laboratory experiments and results of field studies and the link with hypoxia.

Line **No. 524-526**: "This colour zonation is typical for sediments that have been geochemically affected by cable bacteria activity, as seen both in laboratory experiments (Nielsen and Risgaard-Petersen 2015) and at coastal field sites (Sulu-Gambari et al. 2016)"

Line **No. 569-571**: "may act as an additional sediment marker for present or recent cable bacteria activity, both in laboratory experiments and at field sites, also in cases where visual observations are not conclusive."

Line **581-582:** "Macrofaunal activity within natural environments likely counteracts or prevents strong focusing of Fe oxides and associated P within such a thin subsurface layer."

Line **588-590**: "At this site, which is in a region that is subject to seasonal hypoxia (Capet et al. 2013), both bivalves (up to ~7200 ind. m$^{-2}$) and polychaetes (up to ~1700 ind. m$^{-2}$)…"

**Comment #4** Is there any information about the organic carbon content of the sediment and how this changes over the incubation period? I would assume that this is the major driver for the development of biogeochemical zonation.

**Reply:** We now include the organic carbon content of the upper 0-0.5 cm of the sediment at the field site, as determined by Lenstra et al. (2019), in Table 1. We did not determine the change in organic carbon contents during the experiment because, at the typical rates of organic matter degradation expected here, only a small change in organic carbon content would be observed. Hence, we chose to focus on pore water $NH_4^+$ profiles to obtain insight in rates of (anaerobic) organic matter degradation.

**Comment #5** How does the development of the oxic zone, as shown in the experiment, relate to natural variations in coastal sediments ?

**Reply:** The range in $O_2$ penetration observed in the experiment is comparable to that observed in coastal systems with seasonal hypoxia (e.g. Seitaj et al. 2015). We included this in the manuscript: Line **No. 535-538**: "During the experiment, $O_2$ penetration varied within a narrow range and was initially fixed between 1 and 2 mm depth (Fig. 3A), with the layer highly enriched in Fe forming mostly at a depth of 2 mm (Fig. 8A). Such a range in $O_2$ penetration is in accordance with observations in coastal sediments (e.g. Seitaj et al. 2015). The formation of the Fe-enriched layer can be explained by…"

**Comment #6** How does the experiment relate to the development of hypoxia and depletion of oxygen in coastal areas ? The experiment shows the opposite reaction (from anoxic surface layer to an oxygenated layer).

**Reply:** As shown in previous work that we refer to in our manuscript (Seitaj et al. 2015; Sulu-Gambari et al. 2016), cable bacteria can induce formation of Fe and Mn oxides in seasonally hypoxic coastal systems during periods of oxygenated bottom waters in spring. As explained in line **No. 44-46**, the presence of these Fe and Mn oxides can delay the transition towards euxinia by removing hydrogen sulfide. Our experimental setting can be compared to the onset of bottom water re-oxygenation in spring in such seasonally hypoxic environments (i.e. the sediment was stored anoxically, and then exposed to oxygenated water). This allows study of the mineral dissolution and formation reactions in sediments populated by cable bacteria under such conditions, which is the goal of this work.

**Comment #7** line2 121/122: .... with overlying water ....Was this bottom water taken from the site or artificial water, as used for the aquarium?

**Reply:** The 20 cm long cores (filled with 15 cm sediment) were gently submerged in the two aquaria. Hence, the overlying water in the cores at the start of each incubation was the same as the artificial water used in the aquaria.

**Comment #8** line 153: ....... core was place outside the aquarium .....Why was the core taken out ? was the incubation temperature maintained?

**Reply:** The entire experiment was carried out at in a temperature controlled laboratory at 20 °C. We made explicit that the flux incubations also took place at 20 °C, see line **No. 156-157**: "At each time point, one core was placed outside the aquarium at 20 °C..."

**Comment #9** Was the overlying water during the 24-hour incubation for the solute flux measurements stirred to avoid stratification? This could have influenced the flux across the sediment-water interface because stagnant waters lead to an increase of the Diffusive Boundary Layer, which controls the solute exchange.

**Reply:** The overlying water was mixed continuously by bubbling it continuously with air. The airflow was set in such a way that the water was effectively mixed, while the surface layer of the sediment was left undisturbed. We made this more explicit in the methods, see line **No. 156-160**: "At each time point, one core was placed outside the aquarium at 20 °C, and the isolated volume of overlying water in the core was continuously aerated. Potential stratification of the overlying water was prevented by actively bubbling it. Parafilm was wrapped on top of the cores to prevent evaporation."

**Comment #10** Pore water profiles (especially Fig 1a, Fig 2) are very small and it is difficult to recognize the different profiles ($O_2$, pH, $H_2S$) different; graphs should be enlarged.

**Reply:** We enlarged these graphs and we increased the font sizes to improve the readability of the figures.

**References**

[revised manuscript text omitted]

**A**
[Figure]

**Diffusive O₂ Uptake**

[Figure]

**B**

**Current Density**